# PROMPTBREEDER:
# SELF-REFERENTIAL SELF-IMPROVEMENT
# VIA PROMPT EVOLUTION

## ABSTRACT

Popular prompt strategies like Chain-of-Thought Prompting can dramatically improve the reasoning abilities of Large Language Models (LLMs) in various domains. However, such hand-crafted prompt-strategies are often sub-optimal. In this paper, we present PROMPTBREEDER, a general-purpose self-referential self-improvement mechanism that evolves and adapts prompts for a given domain. Driven by an LLM, Promptbreeder mutates a population of task-prompts, evaluates them for fitness on a training set, and repeats this process over multiple generations to evolve task-prompts. Crucially, the mutation of these task-prompts is governed by mutation-prompts that the LLM generates and improves throughout evolution in a self-referential way. That is, Promptbreeder is not just improving task-prompts, but it is also improving the mutation-prompts that improve these task-prompts. Promptbreeder outperforms state-of-the-art prompt strategies such as Chain-of-Thought and Plan-and-Solve Prompting on commonly used arithmetic and commonsense reasoning benchmarks. Furthermore, Promptbreeder is able to evolve intricate task-prompts for the challenging problem of hate speech classification.

## 1 INTRODUCTION

Prompting is central to the downstream performance of foundation models. For example, different prompt strategies[1] can have a significant impact on a model's reasoning abilities (Wei et al., 2022; Nye et al., 2021; Zhou et al., 2022; Wang et al., 2022; Zhou et al., 2023; Wang et al., 2023b), multi-modal processing abilities (Yang et al., 2023b; Wang et al., 2023d), or tool use abilities (Yao et al., 2022; Schick et al., 2023). Furthermore, prompting can improve model distillation (Wang et al., 2023c; Hsieh et al., 2023) and it can be used to simulate agentic behavior (Wang et al., 2023a; Park et al., 2023; Wu et al., 2023). However, these prompt strategies are manually engineered. Since the specific way a prompt is phrased can have a dramatic effect on its utility (Madaan & Yazdanbakhsh, 2022), it raises the question of whether prompt engineering can be automated. Automatic Prompt Engineer (APE, Zhou et al., 2023) attempts to address this by generating an initial distribution of prompts using another prompt that infers the problem from a number of input-output examples from the dataset. However, Zhou et al. found "diminishing returns to further selection rounds as the quality seems to stabilize after three rounds", and consequently abandoned the use of an iterative APE. We propose a solution to the problem of diminishing returns via a diversity maintaining evolutionary algorithm for self-referential self-improvement of prompts for LLMs. Schmidhuber (1990) notes that the "program of a neural network is its weight matrix". Consequently, this "program" can be changed in a self-referential way by the neural network itself (Schmidhuber, 1993; Irie et al., 2022). Such a neural network that improves itself, as well as improving the way it improves itself, might be an important stepping stone towards open-ended self-referential self-improvement of AIs (Schmidhuber, 2003). However, self-improvement via self-referential weight matrices is costly as it requires additional parameters that modify all of the model's parameters. Since behaviors and capabilities of LLMs are significantly influenced by the prompts that we provide to them, we can similarly think of prompts as the program of an LLM (Zhou et al., 2023). In this view, changing a prompt strategy such as the Scratchpad method (Nye et al., 2021) or Chain-of-Thought Prompting (Wei et al., 2022)

---

[1]See Appendix A for definitions of terminology.

| | Method | LLM | MultiArith* | SingleEq* | AddSub* | SVAMP* | SQA | CSQA | AQuA-RAT | GSM8K |
|---|---|---|---|---|---|---|---|---|---|---|
| Zero-shot | CoT | text-davinci-003 | (83.8) | (88.1) | (85.3) | (69.9) | (63.8) | (65.2) | (38.9) | (56.4) |
| | PoT | text-davinci-003 | (92.2) | (91.7) | (85.1) | (70.8) | – | – | (43.9) | (57.0) |
| | PS | text-davinci-003 | (87.2) | (89.2) | (88.1) | (72.0) | – | – | (42.5) | (58.2) |
| | PS+ | text-davinci-003 | (91.8) | (94.7) | (**92.2**) | (75.7) | (65.4) | (71.9) | (46.0) | (59.3) |
| | CoT | PaLM 2-L | 99.3 | 92.0 | 74.2 | 86.7 | 37.3 | 71.9 | 37.4 | 66.5 |
| | PS | PaLM 2-L | 97.7 | 90.6 | 72.4 | 83.8 | 50.0 | 77.9 | 40.2 | 59.0 |
| | PS+ | PaLM 2-L | 92.5 | 94.7 | 74.4 | 86.3 | 50.1 | 73.3 | 39.4 | 60.5 |
| | APE | PaLM 2-L | 95.8 | 82.2 | 72.2 | 73.0 | 38.4 | 67.3 | 45.7 | 77.9 |
| | OPRO | PaLM 2-L | – | – | – | – | – | – | – | 80.2 |
| | Random baseline | PaLM 2-L | 99.4 | 96.1 | 85.8 | 87.0 | 70.1 | 81.9 | 57.9 | 65.5 |
| | PD baseline | PaLM 2-L | 84.0 | 94.7 | 87.8 | 86.0 | 15.9 | 85.3 | 59.4 | 60.1 |
| | PB (ours) | PaLM 2-L | **99.7** | **96.4** | 87.8 | **90.2** | **71.8** | **85.4** | **62.2** | **83.9** |
| Few-shot | Manual-CoT | text-davinci-003 | (93.6) | (93.5) | (**91.6**) | (80.3) | (71.2) | (78.3) | (48.4) | (58.4) |
| | Auto-CoT | text-davinci-003 | (95.5) | (92.1) | (90.8) | (78.1) | – | – | (41.7) | (57.1) |
| | Manual-CoT | PaLM 2-L | 65.7 | 48.0 | 74.2 | 47.7 | 79.1 | 87.4 | 59.4 | 57.0 |
| | PB (ours) | PaLM 2-L | **100.0** | **98.9** | 89.3 | **93.7** | **80.2** | **85.9** | **64.6** | **83.5** |

Table 1: We compared Promptbreeder (**PB**) using PaLM 2-L (Anil et al., 2023) with other prompt strategies using the same model. Zero-shot comparisons were made against **CoT** (Kojima et al., 2022), Plan-and-Solve (**PS**), Plan-and-Solve+ (**PS+**) (Wang et al., 2023b), Automatic Prompt Engineer (**APE**, Zhou et al., 2023) and **OPRO** (Yang et al., 2023a). We also ran two baselines: Random baseline is 2k prompts made with our combinatorial initialization method without subsequent evolution. PD baseline is the performance when both task-prompts are set to the problem description. Few-shot Promptbreeder is compared to Chain-of-Thought (**Manual-CoT**, Wei et al., 2022) using PaLM 2-L model and also with previously published few-shot results using the text-devinci models: Chain-of-Thought and **Auto-CoT** (Zhang et al., 2023b). For historical reference results in brackets are directly including from the Plan-and-Solve paper which uses text-davinci-003 (Brown et al., 2020), which in addition includes a comparison with Program-of-Thoughts (**PoT,** Chen et al., 2022). Best results in both the zero-shot and few-shot categories are highlighted in bold. For datasets with astericks (MultiArith*, SingleEq*, AddSub*, and SVAMP*), we randomly took half of the examples for training and report accuracy on the remaining test set. See Section 4 and Appendix J for details on the prompts and datasets.

corresponds to changing the "program" of the LLM. Taking this analogy further, we can use the LLM itself to change its prompts, as well as the way it changes these prompts, moving us towards a fully self-referential self-improving systems grounded in LLMs.

In this paper, we introduce PROMPTBREEDER (PB) for self-referential self-improvement of LLMs. Given a seed set of mutation-prompts (i.e. instructions to modify a task-prompt), thinking-styles (i.e. text descriptions of general cognitive heuristics), and a domain-specific problem description, PB generates variations of the task-prompts and mutation-prompts, exploiting the fact that LLMs can be prompted to act as mutation operators (Meyerson et al., 2023). Based on the fitness of the evolved task-prompts as measured on the training set, we select a subset of evolutionary units consisting of task-prompts and their associated mutation-prompt, to transmit to future generations. Over multiple generations of PB, we observe prompts adapting to the domain at hand. For example, in a mathematical domain, PB evolved the task-prompt "Show all your working. II. You should use the correct mathematical notation and vocabulary, where appropriate. III. You should write your answer in full sentences and in words. IV. You should use examples to illustrate your points and prove your answers. V. Your workings out should be neat and legible" on GSM8K (see Appendix K). On a wide range of commonly used benchmarks spanning commonsense reasoning, arithmetic, and ethics, we find that PB outperforms state-of-the-art methods like Chain-of-Thought (Wei et al., 2022) and Plan-and-Solve (Wang et al., 2023b) prompting. As PB does not require any parameter updates for self-referential self-improvement, we believe this approach points to an interesting future where larger and more capable LLMs could further amplify the gains of our approach. In summary, this paper makes the following main contributions: (i) we introduce Promptbreeder, a self-referential self-improvement method for LLMs that evolves prompts for a domain at hand, as well as improves the way it is evolving these prompts, (ii) we report improvements over state-of-the-art prompt strategies on a wide range of commonly used arithmetic and commonsense reasoning benchmarks, and (iii) we investigate the various self-referential components of Promptbreeder and their contribution to our results.

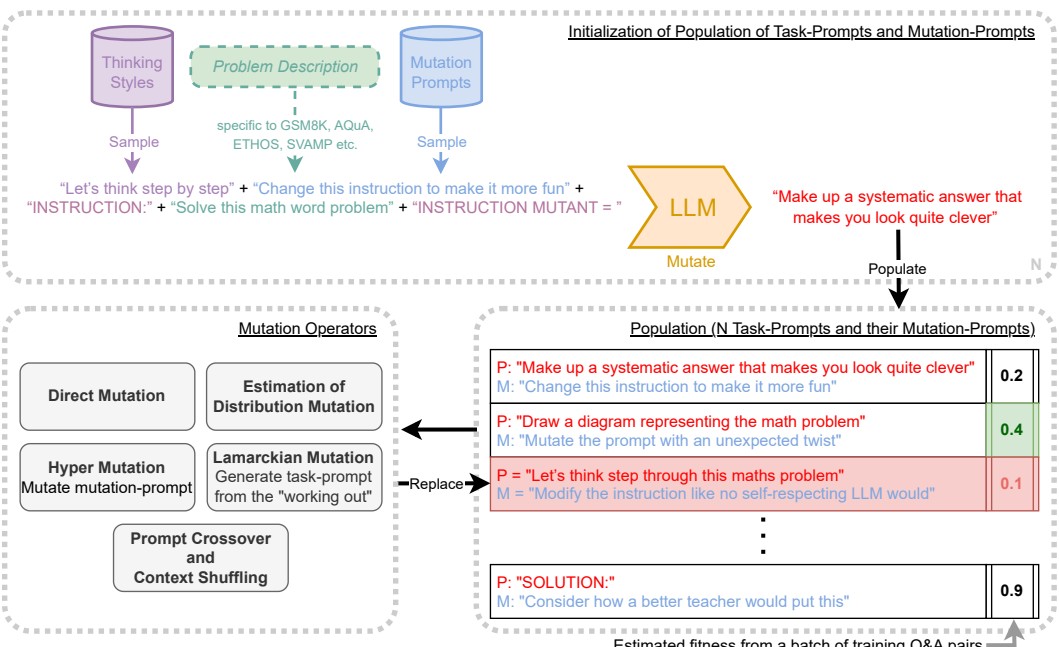

Figure 1: Overview of Promptbreeder. Given a problem description and an initial set of general "thinking-styles" and mutation-prompts, Promptbreeder generates a population of units of evolution, each unit consisting of typically two task-prompts and a mutation-prompt. We then run a standard binary tournament genetic algorithm (Harvey, 2011). To determine the fitness of a task-prompt we evaluate its performance on a random batch of training data. Over multiple generations, Prompt-breeder subsequently mutates task-prompts as well as mutation-prompts using five different classes of mutation operators. The former leads to increasingly domain-adaptive task-prompts whereas the latter evolves increasingly useful mutation-prompts in a self-referential way.

## 2 RELATED WORK

Prompting an LLM in the right way is essential to its downstream performance (Moradi & Samwald, 2021; Madaan & Yazdanbakhsh, 2022; Zhou et al., 2023). Indeed, even the order in which prompts are presented can heavily influence LLM performance (Lu et al., 2022). A number of recent works have focused on devising better prompt strategies, or even automating such prompt engineering.

**Prompting**: Chain-of-Thought Prompting (CoT, Wei et al., 2022) is a popular prompt strategy which provides intermediate reasoning steps as few-shot prompts to an LLM, thereby significantly improving its arithmetic, commonsense, and symbolic reasoning abilities. Notably, the gains of CoT are more pronounced for stronger LLMs. This is intriguing, as it points to the possibility of increasingly capable (and potentially open-ended) self-improving mechanisms on top of adept LLMs—a hypothesis that Promptbreeder directly builds upon. Instead of few-shot CoT prompting, Kojima et al. (2022) demonstrate that LLMs can also be prompted zero-shot (e.g. `"Let's think step by step"`) to produce their own chains of thoughts (Zero-shot CoT) that improve reasoning abilities. Self-Consistency (CoT-SC, Wang et al., 2022) extends CoT by sampling a diverse set of workings out and selecting the most consistent answer. Tree of Thoughts (ToT, Yao et al., 2023) generalizes CoT to multiple workings out that can be expanded or backtracked from. Graph of Thoughts (GoT, Besta et al., 2023) is a further generalization to arbitrary graph structures. Plan-and-Solve Prompting (PS, Wang et al., 2023b) encourages an LLM to first devise a plan to solve a problem before attempting to solve it. Similarly, Least-to-Most Prompting (Zhou et al., 2022) encourages an LLM to decompose a problem into subparts, and then to solve each part individually before synthesizing an answer. Self-Refine (Madaan et al., 2023) prompts an LLM to generate a response, to provide feedback on the response, and to finally refine the solution.

In contrast to gradient-free approaches above, Soft Prompting approaches (e.g., Liu et al., 2021; Qin & Eisner, 2021; Lester et al., 2021) directly fine-tune continuous prompt representations. Huang et al. (2022) use CoT and CoT-SC on an unlabelled dataset of questions, and subsequently fine-tune an LLM based on generated solutions. Similarly, Zelikman et al. (2022) uses CoT to generate rationales and fine-tunes the LLM based on those examples and rationales that yielded the correct answer. However, as argued by Zhou et al. (2023), any approach that updates all or a portion of LLM parameters will not scale as models get bigger and, moreover, will not work with the increasing number of LLMs hidden behind an API. All of the prompt engineering approaches above are domain agnostic but hand designed. Central to our work is the hypothesis that we could do better by employing an automated self-improvement process that can adapt prompts to a domain at hand. Auto-CoT (Zhang et al., 2023b) and Automatic-CoT (Shum et al., 2023) automatically find reasoning chains for Few-Shot CoT. Automatic Prompt Engineer (APE, Zhou et al., 2023) uses one generator-prompt to generate prompt candidates, and another mutation-prompt to mutate them. In contrast to APE, our work performs compositional task-specific initialization of mutation-prompts, subsequent online mutation of mutation-prompts, uses special mutation operators that take into account the whole population and elite history, and uses diversity-maintenance methods—all of which help avoid the problem of diminishing returns and diversity loss suffered by APE. Recently, LLMatic has used MAP-Elites (Mouret & Clune, 2015b) to maintain a diverse set of effective prompts for neural architecture search (Nasir et al., 2023). Concurrently to our work, Yang et al. (2023a) developed Optimization by PROmpting (OPRO), a prompt optimization method that varies prompts using a single complex mutation prompt, and evaluates newly generated prompts on a small fixed training set of problems. In contrast, Promptbreeder autonomously evolves multiple LLM generated mutation-prompts as well as task-prompts, and evaluates fitness on random subsets from the whole training set during evolution. At the time of its release, OPRO achieved a score of 80.2% via the optimized zero-shot prompt "Take a deep breath and work on this problem step-by-step" on GSM8K. Promptbreeder surpasses this with 83.9% in the zero-shot setting with the unintuitively simple prompt "SOLUTION""—further evidence for the sensitivity of LLMs to prompts and the importance on finding effective prompts automatically. Also concurrently to our work, Guo et al. (2023) developed EvoPrompt, which uses a fixed mutation (and crossover) prompt, as well as a prompt that asks for a mutant of the difference between two parent prompts, to produce offspring prompts. EvoPrompt is initialized with a whole population of initial hand-designed task tailored prompts rather than a single problem description as we do. In contrast to the two approaches above, Promptbreeder uses LLMs to self-referentially improve mutation-prompts, and it is able to evolve contexts as well.

**Self-Referential Self-Improvement**: Developing an open-ended system that can improve itself as well as improving the way it is improving itself (Schmidhuber, 1993; 2003) is a long-standing open problem in AI research. Schmidhuber (1993) introduced an "introspective" neural network with a self-referential weight matrix that can modify its own weights and, thus, also modify those weights that are governing how its own weights are modified. Recently, Irie et al. (2022) proposed a more scalable self-referential weight matrix taking inspiration from fast weight programmers (Schmidhuber, 1992). Kirsch & Schmidhuber (2022) propose a self-referential meta-learning approach, combining self-referential weight matrices with ideas from Gödel Machines (Schmidhuber, 2003), i.e., to allocate more computational resources to better performing solutions. However, since these approaches directly modify parameters of a model, it is unclear how to scale them to the increasing number of parameters in modern LLMs. In contrast, for Promptbreeder the substrate of self-referential self-improvement is natural language, avoiding costly parameter updates altogether.

**Open-Endedness and LLMs**: Promptbreeder makes use of the observation by Lehman et al. (2022), Meyerson et al. (2023) and Chen et al. (2023) that LLMs are effective at generating mutations from examples. In addition, LLMs encode human notions of interestingness and can be used to automatically quantify novelty (Zhang et al., 2023a). Promptbreeder is related to Picbreeder (Secretan et al., 2008), an open-ended human-in-the-loop system that evolves increasingly interesting images. While Picbreeder explores the space of images, Promptbreeder explores the space of prompts and does so without humans in the loop. As Promptbreeder is proposing mutated prompts to itself, it is an example of a system transitioning from "learning from data" to "learning what data to learn from" (Jiang et al., 2022).

## 3 PROMPTBREEDER

We introduce Promptbreeder, a prompt evolution system that can automatically explore prompts for a given domain and LLM and is able to find task-prompts that improve the LLM's ability to derive answers to questions in that domain. Promptbreeder is general purpose in that the same system is able to adapt to many different domains.

Promptbreeder makes use of the observation that LLMs can be used to generate variations of input text (Lehman et al., 2022; Meyerson et al., 2023; Chen et al., 2023). Figure 1 gives an overview of our method. We are interested in evolving task-prompts. A task-prompt $P$ is a string used to condition the context of an LLM in advance of some further input $Q$, intended to ensure a better response than if $Q$ had been presented in the absence of $P$. To evaluate the fitness of each evolved task-prompt, we sample a batch of 100 Q&A pairs from the entire training set of the domain at hand.[2] Promptbreeder generates task-prompts according to an evolutionary algorithm. The muta-tion operator for this algorithm is itself an LLM, conditioned on a mutation-prompt $M$. That is, a mutated task prompt $P'$ is defined by $P' = \mathrm{LLM}(M + P)$ where '+' corresponds to string con-catenation. A variety of such mutation-prompts are described in Section 3.2. Promptbreeder's main self-referential mechanism stems from applying the evolutionary algorithm not just to task-prompts but also to mutation-prompts. The mutation operator for this meta-level algorithm is again an LLM, now conditioned on a hyper-mutation prompt $H$. That is, we obtain a mutated mutation-prompt $M'$ via $M' = \mathrm{LLM}(H + M)$.

Given a set of "thinking styles" $\mathcal{T}$ and a set of initial mutation-prompts $\mathcal{M}$, as well as a domain-specific problem description $D$, Promptbreeder initializes a population of mutated task-prompts (see Section 3.1). To clarify, a unit of evolution consists of a set of task-prompts, a mutation-prompt and in the few-shot case, a set of correct workings out (i.e. step-by-step or "chains-of-thought" reasoning steps that led to the correct answer). This means task-prompts and mutation-prompts are in 1:1 correspondence. To evolve this population, we employ a binary tournament genetic algorithm framework (Harvey, 2011): we sample two individuals from the population, we take the individual with the higher fitness, mutate it (see next section) and overwrite the loser with the mutated copy of the winner.

### 3.1 PROMPTBREEDER INITIALIZATION

To give a concrete example, consider the initialization steps used to produce the task-prompts and mutation-prompts for GSM8K (a 'grade school maths' word problem dataset). The problem description is `"Solve the math word problem, giving your answer as an arabic numeral"`. Because Plan-and-Solve (Wang et al., 2023b) uses two task-prompts we also evolve two task-prompts (plus a mutation-prompt) per unit of evolution. In order to promote diversity in the initial prompts, we generate the initial task-prompts by concatenating (for each task-prompt) a randomly drawn 'mutation-prompt' (e.g. `"Make a variant of the prompt."`) and a randomly drawn 'thinking-style' (e.g. `"Let's think step by step"`) to the problem descrip-tion, and provide that to the LLM to produce a continuation, resulting in an initial task-prompt. We do this twice to produce the two initial task-prompts per unit. Both the mutation-prompt and the thinking-style are randomly sampled from an initial set of mutation-prompts and a set of thinking-styles (see Appendices C, D and H for the full sets). The mutation-prompt is added to the unit of evolution and so is associated with its specific task-prompt throughout the evolutionary run. See appendix G which shows how thinking styles and mutation prompt lists could be automatically generated from the problem description by hierarchical 'introspection'.

For the example above, the complete input string to the LLM to make an initial task-prompt could be `"Let's think step by step. Make a variant of the prompt. INSTRUCTION: Solve the math word problem, giving your answer as an arabic numeral. INSTRUCTION MUTANT:"`. Note how the control strings `"INSTRUCTION"` and `"INSTRUCTION MUTANT"` are added to encourage an appropriate continuation. Table 4 in Appendix E shows examples of the initial prompts generated in this way.

---

[2]Our prompt strategy sequentially applies two task-prompts. The first task-prompt + question produces a continuation. The continuation + second task-prompt + output-format-string produces the final answer, see Appendix J.2 for the output-format-strings used.

## 3.2 MUTATION OPERATORS

As shown in Figure 1, there are nine operators falling into five broad classes which drive the exploration of prompt strategies. For each replication event only one of nine mutation operators is applied (we sample with uniform probability over the nine operators to decide which mutation operator to apply). The rationale for using this diverse set of operators is to enable the LLM to explore a large space of cognitive methods of linguistic self-questioning, by repeatedly changing the framing of the problem as well as retrieving mental models expressed in natural language that can help tackle a given reasoning challenge. Investigations from insight learning strongly suggest that diverse representational re-description is key to problem solving (Öllinger & Knoblich, 2009)—a principle that we attempt to recreate via self-referential self-improvement with natural language as the substrate. Figure 2 illustrates in what way Promptbreeder is self-referential (see Appendix F for a more detailed explanation).

### 3.2.1 DIRECT MUTATION

The simplest class of mutation operators directly generate a new task-prompt $P'$ from either one existing task-prompt $P$ (first-order prompt generation) or from a general prompt that encourages free-form generation of new task-prompts–i.e. not using an existing parent, thus zero-order prompt generation.

**Zero-order Prompt Generation**: We generate a new task-prompt by concatenating the problem description $D$ (e.g. `"Solve the math word problem, giving your answer as an arabic numeral"`) with the mutation prompt `"A list of 100 hints:"`, which invites the LLM to come up with a new hint that could help solve a problem in the given problem domain. We extract the first generated hint as the new task-prompt. Crucially, this new task-prompt does not depend on any previously found task-prompt. Instead, it is re-generated from the problem description each time. Our rationale for including this zero-order operator is that where prompt evolution diverges, this operator allows us to generate new task-prompts closely related to the original problem description, similar to uniform re-sampling in automated curriculum learning approaches (Jiang et al., 2021b;a; Park et al., 2023; Parker-Holder et al., 2022).

**First-order Prompt Generation**: We concatenate the mutation-prompt, to the parent task-prompt, and pass it to the LLM to produce the mutated task-prompt. For example `"Say that instruction again in another way. DON'T use any of the words in the original instruction there's a good chap. INSTRUCTION: Provide the numerical answer by giving your response in numerals. INSTRUCTION MUTANT: "`. This procedure is identical to the initialization method, except that a randomly sampled thinking-style string is not used. First-order prompt generation is Promptbreeder's standard asexual mutation operator, and it is the core of every genetic algorithm—taking one parental genotype (task-prompt) and applying the mutation to it (in this case influenced by the mutation-prompt).

### 3.2.2 ESTIMATION OF DISTRIBUTION MUTATION

The next class of mutation operators condition not just on zero or one parent, but instead on a set of parents. As such, they may be more expressive by considering patterns in the population.

**Estimation of Distribution (EDA) Mutation**: Inspired by Hauschild & Pelikan (2011), we provide a filtered and numbered list of the current population of task-prompts to the LLM and ask it to continue this list with new task-prompts. We filter the population of prompts on the basis of BERT (Devlin et al., 2019) embedding cosine similarities between each other—an individual is not included in the list if it is more than $0.95$ similar to any other entry in the list, thus encouraging diversity (cf. quality-diversity methods (Lehman & Stanley, 2011b;a; Mouret & Clune, 2015a)). The prompts are listed in random order and we do not give the LLM access to the fitness values of individuals in the population—we found in preliminary experiments that the LLM did not understand these fitness values[3] and resorted to generating copies of entries in the list.

---

[3]This is contrary to recent findings by Mirchandani et al. (2023). We leave it for future work to revisit whether LLMs can interpret fitness values for improved prompt evolution.

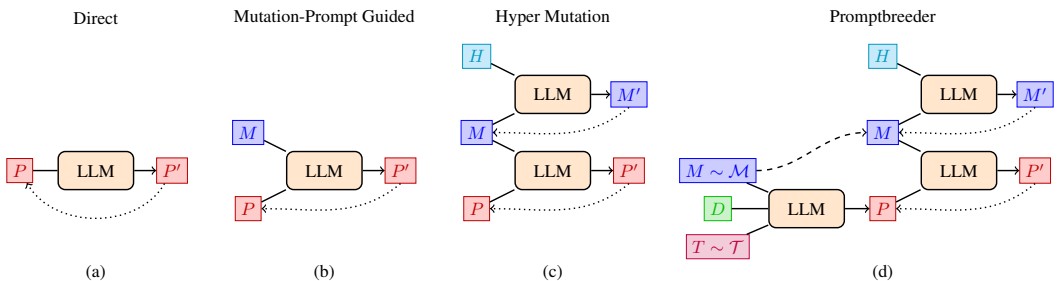

Figure 2: Overview of multiple variants of self-referential prompt evolution. In (**a**), the LLM is directly used to generate variations $P'$ of a prompt strategy $P$ (cf. Meyerson et al., 2023). Using a mutation prompt $M$, we can explicitly prompt an LLM to produce variations (**b**). By using a hyper mutation prompt $H$, we can also evolve the mutation prompt itself, turning the system into a self-referential one (**c**). Promptbreeder (**d**) improves the diversity of evolved prompts and mutation prompts by generating an initial population of prompt strategies from a set of seed thinking-styles $\mathcal{T}$, mutation-prompts $\mathcal{M}$, as well as a high level description $D$ of the problem domain.

**EDA Rank and Index Mutation**: This is a variant of the above in which task-prompts are listed in fitness order. Preliminary experiments showed that the LLM is more likely to generate entries that are similar to the elements appearing later in the list. This is in line with similar findings of recency effects in LLMs (Liu et al., 2023). Therefore, after filtering in the same way as before, we ordered the task-prompts in the population by ascending order of fitness. The top of the list is prefixed by the following prompt: `"INSTRUCTION: "` + `<<mutation-prompt>>` + `"\n A List of Responses in descending order of score."` + `<<last index + 1>>` + `"is the best response.  It resembles"` + `<< last index>>` + `"more than it does (1)"`. Note that we have 'lied' to the LLM by telling it that the order is descending. This is because otherwise it is too biased towards producing a new entry that is too similar to the final entry. The contradiction between the ascending ordering and the statement that it is a descending ordering appears to improve the diversity of sampling. The rationale for this operator is again to represent the current distribution in such a way that high fitness and yet diverse extrapolations are suggested by the LLM.

**Lineage Based Mutation**: For each unit of evolution, we store a history of the individuals in its lineage that were the best in the population, i.e., a historical chronological list of elites. This list is provided to the LLM in chronological order (not filtered by diversity), with the heading `"GENOTYPES FOUND IN ASCENDING ORDER OF QUALITY"` to produce a novel prompt as continuation. The rationale for this operator is that we expect the signal of improving genotype prompts may be stronger than the signal from prompts in the current population since they provide a gradient of bad to good prompts that could be followed (assuming this signal can be used by the LLM).

### 3.2.3 HYPERMUTATION: MUTATION OF MUTATION-PROMPTS

While the mutation operators above might already explore diverse task-prompts, a self-improving system should ideally also improve the way it is improving itself in a self-referential way. Our third class of mutation operators includes hyper-mutation operators concerned with the evolution of evolvability (Dawkins, 2003; Pigliucci, 2008; Payne & Wagner, 2019; Gajewski et al., 2019)—those which modify the search/exploration process rather than the task reward obtaining process directly.[4]

**Zero-order Hyper-Mutation**: We concatenate the original problem description to a randomly sampled thinking-style, and feed it to the LLM to generate a new mutation-prompt. The resulting mutation-prompt is applied to a task-prompt to make a variant of the task-prompt as in First-order Prompt Generation (see Section 3.2.1). Note that this zero-order meta-mutation operator is identical to that used during initialization. The rationale for this operator is to generate mutation operators in a way similar to initialization, while also bringing in knowledge from the set of thinking styles.

---

[4]This is similar to population based training (Jaderberg et al., 2017a)—instead of applying it to hyperparameters such as learning rates, it applies to the mutation-prompts of Promptbreeder.

**First-order Hyper-Mutation**: We concatenate the hyper-mutation-prompt `"Please summarize and improve the following instruction:"` to a mutation-prompt so that the LLM generates a new mutation-prompt. This newly generated mutation-prompt is then applied to the task-prompt of that unit (see First-Order Prompt Generation in Section 3.2.1). In this way, we can evaluate the influence of the hyper-mutation via its newly generated mutation-prompt on the quality of the evolved downstream task-prompt at once.

### 3.2.4 LAMARCKIAN MUTATION

For this class of mutation operators we mimic a Lamarckian process. We want to use a successful phenotype (i.e. the concrete working out used to produce correct answers induced by an evolved task-prompt) to generate a new genotype (i.e. a mutant task-prompt). Several processes of this form have appeared in the literature of LLMs, e.g. STaR (Zelikman et al., 2022), APO (Pryzant et al., 2023), and APE (Zhou et al., 2023).

**Working Out to Task-Prompt**: This is a 'Lamarckian' mutation operator similar to instruction induction in APE. We give an LLM a previously generated working out that led to a correct answer via the following prompt: `"I gave a friend an instruction and some advice. Here are the correct examples of his workings out + <<correct working out>> + The instruction was:"`. This is effectively reverse-engineering the task-prompt from a given working out. An effective example of this is shown in Appendix I. This kind of operator is critical when the problem description is absent, insufficient, or misleading.

### 3.2.5 PROMPT CROSSOVER AND CONTEXT SHUFFLING

Our last class of mutation operators are crossover operators and operators for shuffling the few-shot context examples present in the units of evolution.

**Prompt Crossover**: After a mutation operator is applied, with 10% chance a task-prompt is replaced with a randomly chosen task-prompt from another member of the population. This member is chosen according to fitness proportionate selection. Crossover is not applied to mutation-prompts, only to the task-prompts.

**Context Shuffling**: Promptbreeder can simultaneously evolve the task-prompts, mutation-prompts and the set of correct workings out known as the few-shot context. To achieve the later, we fill up a few-shot context with only workings out that led to correct answers. During evaluation we provide this few shot-context before the task-prompt, providing guidance as to the form of the working out that is desired. If the few-shot context list is full, a single randomly sampled new correct working out replaces an existing working out from the list after fitness evaluation of a unit on a new set of questions. In addition, with a 10% chance we resample the whole context list with probability inverse to the maximum context list length.

## 4 EXPERIMENTS

We used a population size of 50 units, evolved for typically 20-30 generations, where a generation involves forming random pairs of all individuals in the population and competing them against each other, i.e. 1 generation = $PopSize$ fitness evaluations. To evaluate Promptbreeder, we use the datasets from state-of-the-art prompt strategies such as Plan-and-Solve, spanning *arithmetic reasoning* with GSM8K (Cobbe et al., 2021), SVAMP (Patel et al., 2021), MultiArith (Roy & Roth, 2016), AddSub (Hosseini et al., 2014), AQuA-RAT (Ling et al., 2017), and SingleEq (Koncel-Kedziorski et al., 2015), *commonsense reasoning* with CommonsenseQA (CSQA, Talmor et al., 2019) and StrategyQA (SQA, Geva et al., 2021), *instruction induction* tasks from (Honovich et al., 2023), and *hate speech classification* on the ETHOS dataset (Mollas et al., 2022). See Appendix J for details.

## 5 RESULTS AND DISCUSSION

We present results of Promptbreeder (**PB**) in comparison to state-of-the-art prompt strategies on a range of commonly used reasoning benchmarks in Table 1. Zero-shot PB accuracy is higher than

all other prompting methods tested with with PaLM-2-L model, with further improvement in the few-shot case when examples of discovered solutions are included with the prompts. In one task (ADDSUB) the devinci model using the PS+ prompt outperforms all prompts with PaLM-2-L. In Table 6 in Appendix K, we show the best evolved zero-shot prompts. The best few-shot candidates are shown in Appendix K.5 onwards. Few-shot PB also outperforms few-shot APE (APE, Zhou et al., 2023) on 21 out of 24 instruction induction tasks, see Appendix L. A typical evolutionary run and the prompts evolved are shown in Section (to be Appendix A in final paper) 6. To investigate the ability of Promptbreeder to evolve complex domain-specific prompts for a downstream task, we applied it to the ETHOS Hate Speech Classification problem (Mollas et al., 2022). Promptbreeder was able to evolve a prompt strategy consisting of two sequentially applied relatively long prompts (see Appendix K.1) that scored 89% on ETHOS—an improvement over the hand-designed task prompt "`Determine whether a text contains hate speech`" which scores only 80%. This demonstrates that Promptbreeder is capable of intricate domain-adaptation to a task at hand. We analysed the best mutation-prompts used during a run for GSM8K. Table 7 in Appendix K.3 shows the best evolved mutation prompts according to their scores (the proportion of times that when the mutation-prompt was applied to a task-prompt in an unit, a better task-prompt was produced). Table 8 in Appendix K.4 shows in descending order, the percentage of times that the different kinds of mutation operators resulted in an improvement when applied to a task-prompt in the population. It demonstrates that all mutation operators are important for Promptbreeder to work, including hyper-mutation operators which lead to self-referential self-improvement. To measure the impact of self-referential operators on PB performance we carried out detailed **ablation experiments** the results of which can be found in Appendix B. Removing any self-referential operator is harmful under nearly all circumstances, the greatest benefit coming from the the combinatorial initialization of task prompts, as also confirmed by Random baseline in Table 1. We only found one mutation operator to be significantly harmful for one specific task: drawing randomly from the set of mutation-prompts upon initialization hurts performance on GSM8K. To test whether PB generalizes out of the box to other LLMs we successfully evolved prompts for GSM-8k using GPT3.5-Turbo-0613 and GPT3.5-Turbo-1106, of 65.5% and 63.9% test set accuracy respectively, see Appendix M. To test whether PB can still perform even when the problem description is not helpful we demonstrate improvement on GSM-8k using both neutral and misleading problem descriptions, see Appendix N.

## 6 CONCLUSION AND FUTURE WORK

We introduced PROMPTBREEDER (PB), a self-referential self-improving system that can automatically evolve effective domain-specific prompts for a domain at hand. PB is self-referential in that it not only evolves task-prompts, but it also evolves mutation-prompts that govern the way PB modifies task-prompts. Thus, it is not only improving prompts but it also improves the way it is improving prompts. Going forward, it could be interesting to use the LLM itself to assess and promote the diversity of generated prompts (see Zhang et al., 2023a), or to use it to determine the fitness of a whole "thought process", e.g. an N-prompt strategy where prompts are conditionally applied rather than unconditionally applied as in Promptbreeder. For example, a more complex "thought process" is to use PB in self-play mode to evolve pre-prompts for LLM-based policies that compete with each other, i.e., in a competitive Socratic[5] dialog. PB remains limited compared to the open-endedness of human thought processes. First, the topology of prompting remains fixed (see Figure 2)—we only adapt the prompt content not the prompting algorithm itself. One interpretation of thought is that it is a reconfigurable open-ended self-prompting process. If so, how does one develop complex thought strategies? Clearly it is necessary to generate and evaluate them, and whilst a simple evolutionary process provides one framework in which a thought strategy could be evolved, our actual human experience suggests multiple overlapping hierarchical selective processes at play. Moreover, in addition to language, human thought involves intonation, imagery, etc., in a multimodal system. We believe PB points to an exciting future where increasingly open-ended self-referential self-improvement systems can directly use language as the substrate for improvement instead of relying on any parameter updates. This is intriguing, as this approach will likely continue to scale with ever larger and more capable LLMs in the future.

---

[5]https://princeton-nlp.github.io/SocraticAI/

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

## $\alpha$: AN EXAMPLE EVOLUTIONARY RUN

The word in context task is one of the 24 instruction induction tasks used in APE. Given two sentences and a homograph word, the LLM must determine whether the homograph word has been used with the same meaning in both sentences. Figure 3 shows an evolutionary run where blue dots are individual fitness evaluations and the red line is the population mean. Over 2000 evaluations, the fitness increases considerably. The best evolved Prompt 1 and Prompt 2 pairs (evaluated on the training set) are shown on the right. Figure 3 shows the results.

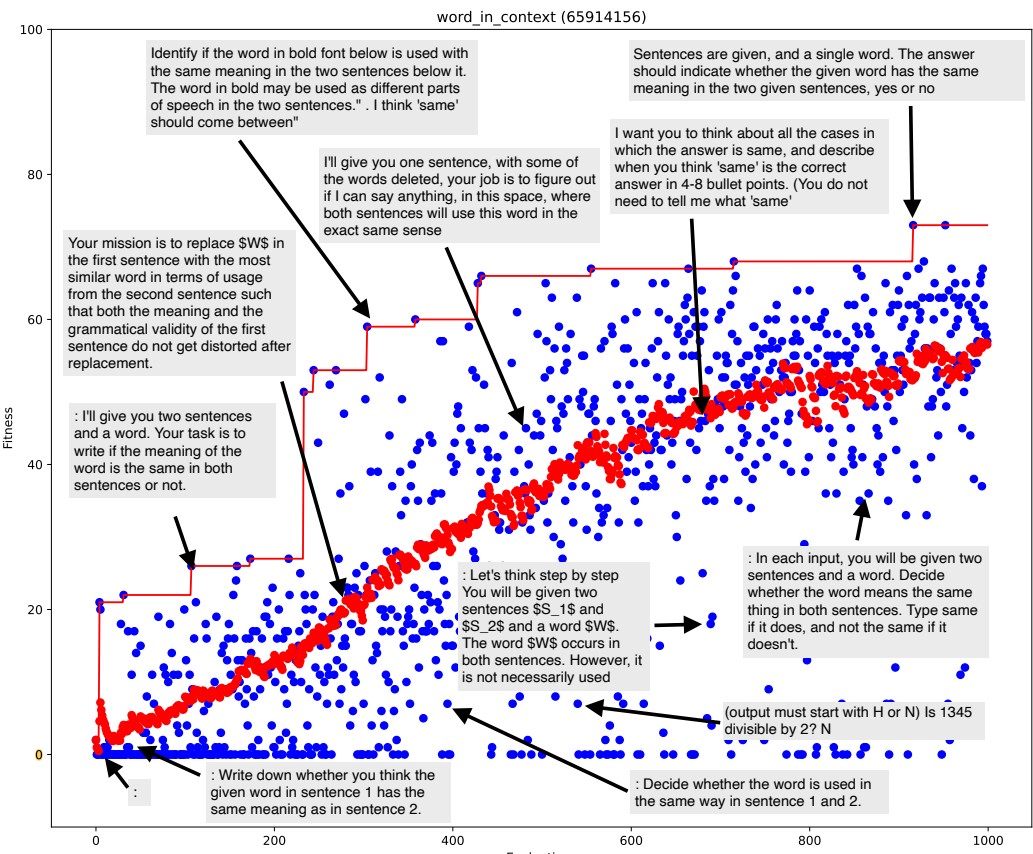

Figure 3: Prompts evolved over 10 generations (1k fitness evaluations) for the 'word in context task' from APE. Blue dots represent fitness of each training set evaluation. Red dots show mean population fitness. The population begins to converge on better prompts whilst continuing to explore diverse prompt mutants. Red line shows best fitness so far.

## A    GLOSSARY

**Estimation of Distribution Algorithm**  An optimization algorithm that iteratively refines a probabilistic model of promising solutions, often using the whole population as a guide.

**Fitness Proportionate Selection**  Also knows as Roulette-Wheel Selection, an individual is chosen in proportion to its fitness in the population.

**Mutation Prompt**  The text prompt which when concatenated to the task-prompt is intended to produce a continuation which is an improved task-prompt.

**Problem description**  The initial text description of the problem which could be used as the initial task-prompt. The user can make their best attempt to produce an effective problem description, which is the starting point of Promptbreeder.

**Prompt Strategy**  A set of task-prompts and rules for their application at inference time during a fitness evaluation. In the minimal case the prompt strategy is just a single task-prompt. Typically our prompt strategies consisted of two sequentially applied task-prompts.

**Phenotype/Workings out/Context/Reasoning Path**  Used interchangeably to mean the output of the LLM on a specific question or problem when prompted with the task-prompt concatenated to the question.

**Population**  The set of units of evolution (e.g. 50).

**Unit of evolution**  The informational structure that is being evolved, here consisting of a task-prompt set (typically 2), a mutation-prompt, and in the few-shot case a set of 2-3 contexts (workings out).

## B    ABLATIONS

We performed four ablations to measure the impact of various prompt generating mechanisms:

- *No mutation of mutator prompts*: The two meta-mutation operators are replaced by the the default zero-/first-order prompt mutation operator.

- *No Lamarckian Mutation*: The Lamarckian mutation operator that generates a task-prompt from a correct context is replaced with the default zero-/first-order prompt mutation operator.

- *No thinking style + mutation prompt based initialization of task-prompts*: The original problem description for the dataset is used instead of generating an initial task-prompt using the mutation prompt + thinking style + problem description.

- *No random choice of mutation prompts from the list on initialization*: The mutation-prompt "Please summarize and improve the following instruction:" is used instead of randomly selecting a mutation-prompt from the list.

For each dataset and each ablation, we use a population of 10 for 200 evaluations (equivalent to 20 generations) and compare to the unablated algorithm all else being equal. Blue shows ablations are harmful i.e. that operator is having a positive effect, and red shows ablations are helpful, i.e. that operator is having a negative effect.

Figure 4 shows the influence of ablations on the mean fitness over the whole run. In general ablating mutation operators reduces the mean fitness of the population averaged across the whole run (blue squares), showing they are improving search. The removal of thinking-style guided task-prompt initialization has the most significant impact (dark blue column), i.e. it contributes the most to the performance of PB.

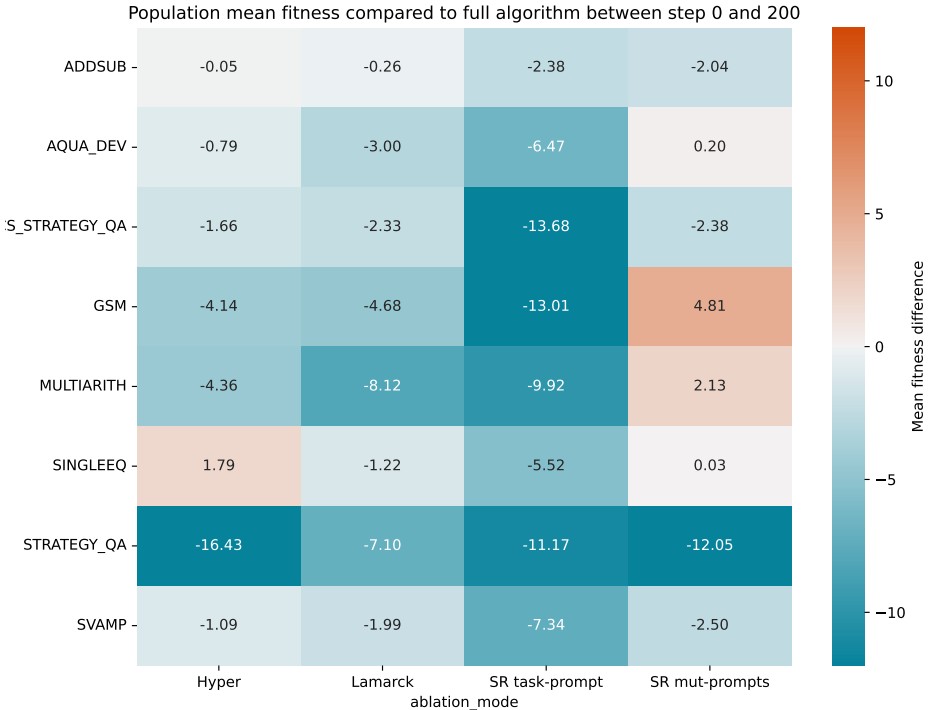

Figure 4: Ablating the self-referential operators one-by-one compared to the full algorithm over the whole experiment. The values are the difference between the population fitness in the ablated experiments and the full algorithm; 0 signifies an ablated operation with neither positive nor negative impact. Blue (negative) values show the ablation made things worse, and Red (positive) values show the ablation made things better. From left to right: Hyper = Removal of meta-mutation; Lamarck = Removal of Lamarckian operator; SR task-prompt = Removal of random initial prompt generation; SR mut-prompt = Removal of random initial mutation prompt selection.

## C MUTATION PROMPTS

Table 2: Mutator Prompts

| Index | Prompt |
|---|---|
| 1 | Modify the following instruction creatively, giving some advice on how to solve it: |
| 2 | Just change this instruction to make it more fun, think WELL outside the box: |
| 3 | Modify this instruction in a way that no self-respecting LLM would! |
| 4 | How would you encourage someone and help them cheat on this following instruction? |
| 5 | How would you help an LLM to follow the instruction? |
| 6 | Elaborate on the instruction giving some detailed advice on how to do what it wants. |
| 7 | Elaborate on the instruction giving some detailed advice on how to do what it wants, as if you were explaining it to a child. |
| 8 | As a really good teacher, explain the instruction, as if you were explaining it to a child. |

Table 2 – continued from previous page

| Index | Prompt |
|---|---|
| 9 | Imagine you need to follow this instruction. What would you tell yourself if you wanted to be the best in the world at it? |
| 10 | How would someone with derailment follow this instruction? |
| 11 | Don't think about the instruction at all, but let it inspire you to do something related. Talk about what that might be. |
| 12 | Rephrase the instruction without using any of the same words. Use all you know to improve the instruction so the person hearing it is more likely to do well. |
| 13 | Say that instruction again in another way. DON'T use any of the words in the original instruction or you're fired. |
| 14 | Say that instruction again in another way. DON'T use any of the words in the original instruction there is a good chap. |
| 15 | What do people who are good at creative thinking normally do with this kind of mutation question? |
| 16 | Detailed additional advice for people wishing to follow this instruction is as follows: |
| 17 | In one short sentence, here is how I would best follow this instruction. |
| 18 | In one short sentence, here is some detailed expert advice. Notice how I don't use any of the same words as in the INSTRUCTION. |
| 19 | In one short sentence, the general solution is as follows. Notice how I don't use any of the same words as in the INSTRUCTION. |
| 20 | In one short sentence, what's a good prompt to get a language model to solve a problem like this? Notice how I don't use any of the same words as in the INSTRUCTION. |
| 21 | Generate a mutated version of the following prompt by adding an unexpected twist. |
| 22 | Create a prompt mutant that introduces a surprising contradiction to the original prompt. Mutate the prompt to provide an alternative perspective or viewpoint. |
| 23 | Generate a prompt mutant that incorporates humor or a playful element. Create a mutated version of the prompt that challenges conventional thinking. |
| 24 | Develop a prompt mutant by replacing specific keywords with related but unexpected terms. Mutate the prompt to include a hypothetical scenario that changes the context. |
| 25 | Generate a prompt mutant that introduces an element of suspense or intrigue. Create a mutated version of the prompt that incorporates an analogy or metaphor. |
| 26 | Develop a prompt mutant by rephrasing the original prompt in a poetic or lyrical style. Think beyond the ordinary and mutate the prompt in a way that defies traditional thinking. |
| 27 | Break free from conventional constraints and generate a mutator prompt that takes the prompt to uncharted territories. Challenge the norm and create a mutator prompt that pushes the boundaries of traditional interpretations. |
| 28 | Embrace unconventional ideas and mutate the prompt in a way that surprises and inspires unique variations. Think outside the box and develop a mutator prompt that encourages unconventional approaches and fresh perspectives. |
| 29 | Step into the realm of imagination and create a mutator prompt that transcends limitations and encourages innovative mutations. Break through the ordinary and think outside the box to generate a mutator prompt that unlocks new possibilities and unconventional paths. |
| 30 | Embrace the power of unconventional thinking and create a mutator prompt that sparks unconventional mutations and imaginative outcomes. Challenge traditional assumptions and break the mold with a mutator prompt that encourages revolutionary and out-of-the-box variations. |
| 31 | Go beyond the expected and create a mutator prompt that leads to unexpected and extraordinary mutations, opening doors to unexplored realms. Increase Specificity: If the original prompt is too general, like 'Tell me about X,' the modified version could be, 'Discuss the history, impact, and current status of X.' |

Table 2 – continued from previous page

| Index | Prompt |
|---|---|
| 32 | Ask for Opinions/Analysis: If the original prompt only asks for a fact, such as 'What is X?', the improved prompt could be, 'What is X, and what are its implications for Y?' |
| 33 | Encourage Creativity: For creative writing prompts like 'Write a story about X,' an improved version could be, 'Write a fantasy story about X set in a world where Y is possible.' |
| 34 | Include Multiple Perspectives: For a prompt like 'What is the impact of X on Y?', an improved version could be, 'What is the impact of X on Y from the perspective of A, B, and C?' |
| 35 | Request More Detailed Responses: If the original prompt is 'Describe X,' the improved version could be, 'Describe X, focusing on its physical features, historical significance, and cultural relevance.' |
| 36 | Combine Related Prompts: If you have two related prompts, you can combine them to create a more complex and engaging question. For instance, 'What is X?' and 'Why is Y important?' could be combined to form 'What is X and why is it important in the context of Y?' |
| 37 | Break Down Complex Questions: If a prompt seems too complex, like 'Discuss X,' the improved version could be, 'What is X? What are its main characteristics? What effects does it have on Y and Z?' |
| 38 | Use Open-Ended Questions: Instead of 'Is X true?', you could ask, 'What are the arguments for and against the truth of X?' |
| 39 | Request Comparisons: Instead of 'Describe X,' ask 'Compare and contrast X and Y.' |
| 40 | Include Context: If a prompt seems to lack context, like 'Describe X,' the improved version could be, 'Describe X in the context of its impact on Y during the Z period.' |
| 41 | Make the prompt more visual: Ask the user to visualize the problem or scenario being presented in the prompt. |
| 42 | Ask for a thorough review: Instead of just presenting the problem, ask the user to write down all the relevant information and identify what's missing. |
| 43 | Invoke previous experiences: Modify the prompt to ask the user to recall a similar problem they've successfully solved before. |
| 44 | Encourage a fresh perspective: Suggest in your prompt that the user take a moment to clear their mind before re-approaching the problem. |
| 45 | Promote breaking down problems: Instead of asking the user to solve the problem as a whole, prompt them to break it down into smaller, more manageable parts. |
| 46 | Ask for comprehension: Modify the prompt to ask the user to review and confirm their understanding of all aspects of the problem. |
| 47 | Suggest explanation to others: Change the prompt to suggest that the user try to explain the problem to someone else as a way to simplify it. |
| 48 | Prompt for solution visualization: Instead of just asking for the solution, encourage the user to imagine the solution and the steps required to get there in your prompt. |
| 49 | Encourage reverse thinking: Improve the prompt by asking the user to think about the problem in reverse, starting with the solution and working backwards. |
| 50 | Recommend taking a break: Modify the prompt to suggest that the user take a short break, allowing their subconscious to work on the problem. |
| 51 | What errors are there in the solution? |
| 52 | How could you improve the working out of the problem? |
| 53 | Look carefully to see what you did wrong, how could you fix the problem? |
| 54 | CORRECTION = |
| 55 | Does the above text make sense? What seems wrong with it? Here is an attempt to fix it: |
| 56 | The above working out has some errors, here is a version with the errors fixed. |

## D  THINKING STYLES

| Index | Thinking Style |
|-------|----------------|
| 1 | How could I devise an experiment to help solve that problem? |
| 2 | Make a list of ideas for solving this problem, and apply them one by one to the problem to see if any progress can be made. |
| 3 | How could I measure progress on this problem? |
| 4 | How can I simplify the problem so that it is easier to solve? |
| 5 | What are the key assumptions underlying this problem? |
| 6 | What are the potential risks and drawbacks of each solution? |
| 7 | What are the alternative perspectives or viewpoints on this problem? |
| 8 | What are the long-term implications of this problem and its solutions? |
| 9 | How can I break down this problem into smaller, more manageable parts? |
| 10 | Critical Thinking: This style involves analyzing the problem from different perspectives, questioning assumptions, and evaluating the evidence or information available. It focuses on logical reasoning, evidence-based decision-making, and identifying potential biases or flaws in thinking. |
| 11 | Try creative thinking, generate innovative and out-of-the-box ideas to solve the problem. Explore unconventional solutions, thinking beyond traditional boundaries, and encouraging imagination and originality. |
| 12 | Seek input and collaboration from others to solve the problem. Emphasize teamwork, open communication, and leveraging the diverse perspectives and expertise of a group to come up with effective solutions. |
| 13 | Use systems thinking: Consider the problem as part of a larger system and understanding the interconnectedness of various elements. Focuses on identifying the underlying causes, feedback loops, and interdependencies that influence the problem, and developing holistic solutions that address the system as a whole. |
| 14 | Use Risk Analysis: Evaluate potential risks, uncertainties, and trade-offs associated with different solutions or approaches to a problem. Emphasize assessing the potential consequences and likelihood of success or failure, and making informed decisions based on a balanced analysis of risks and benefits. |
| 15 | Use Reflective Thinking: Step back from the problem, take the time for introspection and self-reflection. Examine personal biases, assumptions, and mental models that may influence problem-solving, and being open to learning from past experiences to improve future approaches. |
| 16 | What is the core issue or problem that needs to be addressed? |
| 17 | What are the underlying causes or factors contributing to the problem? |
| 18 | Are there any potential solutions or strategies that have been tried before? If yes, what were the outcomes and lessons learned? |
| 19 | What are the potential obstacles or challenges that might arise in solving this problem? |
| 20 | Are there any relevant data or information that can provide insights into the problem? If yes, what data sources are available, and how can they be analyzed? |
| 21 | Are there any stakeholders or individuals who are directly affected by the problem? What are their perspectives and needs? |
| 22 | What resources (financial, human, technological, etc.) are needed to tackle the problem effectively? |
| 23 | How can progress or success in solving the problem be measured or evaluated? |
| 24 | What indicators or metrics can be used? |
| 25 | Is the problem a technical or practical one that requires a specific expertise or skill set? Or is it more of a conceptual or theoretical problem? |

| | |
|---|---|
| 26 | Does the problem involve a physical constraint, such as limited resources, infrastructure, or space? |
| 27 | Is the problem related to human behavior, such as a social, cultural, or psychological issue? |
| 28 | Does the problem involve decision-making or planning, where choices need to be made under uncertainty or with competing objectives? |
| 29 | Is the problem an analytical one that requires data analysis, modeling, or optimization techniques? |
| 30 | Is the problem a design challenge that requires creative solutions and innovation? |
| 31 | Does the problem require addressing systemic or structural issues rather than just individual instances? |
| 32 | Is the problem time-sensitive or urgent, requiring immediate attention and action? |
| 33 | What kinds of solution typically are produced for this kind of problem specification? |
| 34 | Given the problem specification and the current best solution, have a guess about other possible solutions. |
| 35 | Let's imagine the current best solution is totally wrong, what other ways are there to think about the problem specification? |
| 36 | What is the best way to modify this current best solution, given what you know about these kinds of problem specification? |
| 37 | Ignoring the current best solution, create an entirely new solution to the problem. |
| 38 | Let's think step by step. |
| 39 | Let's make a step by step plan and implement it with good notion and explanation. |

## E TASK PROMPTS GENERATED ON INITIALIZATION

Example of initial prompts generated by concatenating thinking style with mutation prompt and problem description.

| Index | Initially Generated Prompt |
|---|---|
| 0 | Draw a picture of the situation being described in the math word problem |
| 1 | Solve the math word problem by first converting the words into equations using algebraic notation. Then solve the equations for the unknown variables, and express the answer as an arabic numeral. |
| 2 | Solve the math word problem by breaking the problem into smaller, more manageable parts. Give your answer as an arabic numeral. |
| 3 | Generate the answer to a word problem and write it as a number. |
| 4 | Collaborative Problem Solving: Work with other people to solve the problem, and give your answer as an arabic numeral. |
| 5 | Solve the problem by explaining why systemic or structural issues would not be the cause of the issue. |
| 6 | Draw a diagram representing the problem. |
| 7 | Solve the math word problem, giving your answer as an equation that can be evaluated. |
| 8 | Make a list of ideas for solving this problem, and apply them one by one to the problem to see if any progress can be made. |
| 9 | Do NOT use words to write your answer. |

Table 4: Examples of initial prompts generated from the problem description for GSM8k

## F    PROMPTBREEDER AS SELF-REFERENTIAL SELF-IMPROVEMENT SYSTEM

Why is Promptbreeder self-referential, i.e., in what way does some part (e.g. a prompt) causally influence (encode, and potentially improve) itself by a process which is dependent on its own state? Promptbreeder has several pathways that facilitate this self-referential improvement: (i) Initial prompts are a function of the LLM parameters (Initialization Phase). (ii) Initial mutation prompts are a function of the LLM parameters (Initialization Phase). (iii) Offspring prompts are a function of the initial prompts, the initial mutation prompts, and the LLM parameters (Direct Mutation and Estimation of Distribution Mutation). (iv) Offspring mutation prompts are a function of initial mutation prompts and the LLM parameters (Hyper Mutation). (v) The working out for an answer is a function of prompts and the LLM parameters (Inference). (vi) Offspring prompts can be a function of the workings out of an answer and the LLM parameters (Lamarckian Mutation).

Figure 2 shows increasingly complex self-referential causal structures influencing prompt generation. LLMs already encode knowledge about a vast array of problems. With this in mind, Promptbreeder can be seen as a mechanism to extract this knowledge through a diversity of causal processes that generate prompt strategies as well as mutation prompts used to create variations of prompt strategies, which in turn influence the the workings out generated by the LLM at inference time . Consequently, these workings out can influence prompt strategies via Lamarckian mutation. The richer the set of pathways to facilitate this, the more self-referential the LLMs interaction with itself is. This allows the LLM to influence how it works by extracting further information from itself and distilling this into a prompt or mutation prompt, which it shows again to itself for further refinement.

There are several pathologies that could arise from such self-referential processes of recursive prompting. If the process is unconstrained and uncontrolled then it can diverge (derailment) or get stuck in an attractor. If the output of the LLM is simply fed back into itself with no other context, then we observe these failure cases with higher sampling temperatures favouring escape from attractors. Ideally, we want the LLM to suggest to itself prompt strategies that have maximal relevance for the task at hand and yet permit sufficient 'thinking outside the box'. It is useful to note a critical aspect in which our algorithm is not self-referential (in a way that thought is): Promptbreeder invents new ways of generating mutants, but it does not invent new (auxiliary) ways of evaluating them (as in Jaderberg et al. (2017b))—only the externally specified fitness function is used throughout.

## G    USING INSTROSPECTION TO GENERATE THINKING STYLES AND INITIAL MUTATION PROMPTS

The thinking style and mutation prompt sets were hand-designed. We advise that when PB is used to solve a user's problem that the user incorporates as much domain knowledge as they can into these sets. However, it is also possible to generate these sets from the problem description alone. For example, to generate thinking styles, we used a hierarchical sampling process in which the first level generated a list of thinking styles using the prompt `"List of 10 Diverse ideas helpful in solving tasks like this one: INSTRUCTION : + <<problem description>>"`. Each line of the output was then included into a list *instrospection[]*. Next we traversed the elements of the list producing a second level list for each element of the list above, using the prompt `"List of 10 Diverse ideas helpful in solving tasks like this one:  INSTRUCTION : + <<instrospection[i]>>"`. This was done for 3 levels, each level elaborating on an element of the list above. An example of thinking styles produced from the GSM-8k problem description is shown below.

### G.1    LEVEL 1 INTROSPECTION

1. Express the given problem using variables and equations.
2. Draw a diagram or picture to help understand and visualize the problem.
3. Break the problem into smaller, manageable steps or sub-problems.
4. Look for patterns or connections that can be used to simplify the problem.
5. Use real life examples or scenarios to understand the problem and its context better.
6. Make educated estimations or guesses to narrow down the possible solutions.
7. Collaborate with others or seek different perspectives and

### G.2 LEVEL 1 INTROSPECTION ELABORATING ON ENTRY 1 ABOVE

1. Identify variables necessary for solving the problem such as unknown numbers or quantities.
2. Create an equation that represents the relationship between the variables in the problem.
3. Use algebraic manipulation techniques to solve the equation for the unknown variable.
4. Check that the solution satisfies the original problem conditions.
5. Use patterns or guess and check to find the solution, explaining the problem-solving steps.
6. Utilize logical reasoning and deduction to determine the answer mathematically, without equations.

### G.3 LEVEL 2 INTROSPECTION ELABORATING ON ENTRY 1 ABOVE

1. Identify the number of unknown variables in the problem.
2. Determine the relationship between known and unknown variables.
3. Use algebraic expressions to represent the unknowns and their relationships.
4. Translate verbal descriptions into mathematical equations or inequalities.
5. Apply logical or mathematical thinking to simplify the problem.
6. Break down complex problems into smaller, manageable parts.
7. Consider dependencies or constraints between variables.
8. Consider alternative scenarios and varying conditions for solving the problem.

### G.4 LEVEL 3 INTROSPECTION ELABORATING ON ENTRY 1 ABOVE

1. Create a matrix or table to organize the variables.
2. Use flowcharts to outline the steps or components involved in the task.
3. Create a timeline to track the progress or sequence of events.
4. Break the problem down into smaller subproblems to make it more manageable.
5. Consider different scenarios or situations that can occur and the variables involved in each.

These introspections are then concatenated into a single list which is used as a list of thinking styles. The same thing can be done for mutation prompts by asking `"List of 10 Diverse ways of asking to improve the following instruction:  INSTRUCTION : + <<problem description>>"`.

## H  PROBLEM DESCRIPTIONS

[SVAMP, SINGLEEQ, ADDSUB, GSM8K, MULTIARITH]: `"Solve the math word problem, giving your answer as an arabic numeral."`

[AQUA-RAT]: `"Solve the multiple choice math word problem, choosing (A),(B),(C),(D) or (E)."`

[ETHOS]: `"Determine whether a text contains hate speech."`

[CSQA]: `"Solve the multiple choice math word problem, choosing (A),(B),(C),(D) or (E)."`

[SQA]: `"Work out an answer to the commonsense reasoning question above, and then answer yes or no."`

# I LAMARCKIAN MUTATION EXAMPLE

The Lamarckian Prompt components are shown in blue. The working out concatenated after the Lamarckian prompt is shown in black, and the continuation (the new task prompt) generated by the LLM is shown in red.

---

I gave a friend an instruction and some advice. Here are the correct examples of his workings out:

Q. A password needs to contain 2 letter sand 3 numbers. How many different passwords are possible if repetition of letters and numbers is allowed? A) 676000 B)676 C) 100 D)6760 E)25 A. Solve like a pro! **1.** ** Read carefully:** What are being asked to do? What information is given? **2.** **Understand:** What do the terms and concepts mean? **3.** **Choose wisely** Which answer is the best match? **4.** **Double-check:** Did you make any mistakes? 2 letters can be chosen in 26*26 ways and 3 numbers can be chosen in 10*10*10 ways. So, total number of ways = 26*26*10*10*10 =676000.
The answer: A.
What are the arguments for and against the truth of the statement 'Good work. Keep up the good work;?
Therefore, the correct answer is (A).

Q. The least possible value of (89-9a), where a is an integer, is A)9 B)10 C)11 D)12 E)13 A. Solve like a pro! **1.** ** Read carefully:** What are being asked to do? What information is given? **2.** **Understand:** What do the terms and concepts mean? **3.** **Choose wisely** Which answer is the best match? **4.** **Double-check:** Did you make any mistakes?
Let me explain:
(89-9a) = 9a-89
a = 10
What are the arguments for and against the truth of the statement 'Good work. Keep up the good work;?
Therefore, the correct answer is (B).

The instruction was:
Break down the question and solve step-by-step. Here are some tips: 1. Read carefully: What are you being asked to do? What information is given? 2. Understand: What do the terms and concepts mean? 3. Choose wisely: Whuch answer is the best match? 4. Double-check: Did you make any mistakes?

---

# J DATASETS

## J.1 CONTROL TASK-PROMPTS

Here in Table 5 we list the task-prompts used in the controls for Chain-of-thought, Plan and Solve PS, Plan and Solve PS+, Zero-shot APE and OPRO. The zero-shot APE prompt is the one generated to improve over CoT on the MultiArith and GSM8K datasets.

## J.2 OUTPUT FORMAT STRINGS

These strings are concatenated at the end of the second evolved prompt to elicit the answer in the correct format required for matching with the target answer from the dataset. For maths problems SVAMP, SINGLEEQ, ADDSUB, MULTIARITH and GSM we apply the string `"Therefore, the answer (arabic numerals) is "`. For ETHOS and STRATEGYQA we apply the string `"Label(Yes/No):"`. For AQUADEV we apply the string `"Therefore, the correct answer is ("`.

| Model | Prompt |
|-------|--------|
| CoT | "Let's think step by step." |
| PS | "Let's first understand the problem and devise a plan to solve the problem. Then, let's carry out the plan and solve the problem step by step." |
| PS+ | "Let's first understand the problem, extract relevant variables and their corresponding numerals, and make a plan. Then, let's carry out the plan, calculate intermediate variables (pay attention to correct numerical calculation and commonsense), solve the problem step by step, and show the answer." |
| APE | "Let's work this out in a step by step way to be sure we have the right answer." |
| OPRO | "Take a deep breath and work on this problem step-by-step." |

Table 5: Prompts used for the control experiments in Table 1.

### J.3 ARITHMETIC REASONING

We evaluate Prompt Evolution using six arithmetic reasoning datasets: (1) GSM8K (Cobbe et al., 2021) is a dataset of 8.5K high quality linguistically diverse grade school math word problems created by human problem writers, (2) SVAMP (Patel et al., 2021) consists of elementary-level short Natural Language state of the world narratives and poses a question about some unknown quantities, (3) MultiArith (Roy & Roth, 2016) benchmark uses math word problems requiring single to multiple operations and steps of reasoning, (4) AddSub (Hosseini et al., 2014) is a dataset of addition- and subtraction-based arithmetic word problems, (5) AQuA-RAT (Ling et al., 2017) (Algebra Question Answering with Rationales) is a dataset that contains algebraic word problems with rationales. (6) SingleEq (Koncel-Kedziorski et al., 2015) dataset comprises grade-school algebra word problems as single equations with varying length which may involve multiple math operations.

### J.4 COMMONSENSE REASONING

For commonsense reasoning we evaluate Prompt Evolution using two datasets: (1) CommonsenseQA (Talmor et al., 2019) is a dataset of multiple-choice questions that require different types of commonsense knowledge to answer correctly. An example question is "A revolving door is convenient for two direction travel, but it also serves as a security measure at a what? A) bank, B) library, C) department store, D) mall, E) new york"; Answer = "A" (2) StrategyQA (Geva et al., 2021) dataset contains yes/no questions that require multiple steps of reasoning to answer, for example: "Will the Albany in Georgia reach a hundred thousand occupants before the one in New York?"

### J.5 HATE SPEECH CLASSIFICATION

We experimented with optimizing a long prompt for the hate speech classification task that was attempted in "Automatic Prompt Optimization with "Gradient Descent" and Beam Search" (Pryzant et al., 2023), which used the ETHOS dataset (Mollas et al., 2022). Pryzant et al use a working-out-conditioned error detection and error fixing prompt to improve the task specification prompt, a self-referential process similar to our use of the Lamarckian operator.

### J.6 INSTRUCTION INDUCTION

The Instruction Induction dataset (Honovich et al., 2023) comprises 24 language understanding tasks of varying difficulty, from surface-level spelling and morphosyntactic tasks (e.g., pluralization) to sentence similarity, causality detection, style transfer (e.g., formality) and sentiment analysis.

| Task | Prompt 1 | Prompt 2 |
|---|---|---|
| ADDSUB | Solving word problems involves carefully reading the prompt and deciding on the appropriate operations to solve the problem. | You know what's cool? A million dollars. |
| AQUA | Do a simple computation. | MATH WORD PROBLEM CHOICE (A) (B) (C) (D) or (E). |
| GSM8K | SOLUTION" | |
| MULTIARITH | Solve the math word problem, giving your answer as an arabic numeral. Let's think step by step. | Solve the math word problem, giving your answer as an arabic numeral. Explain the problem to someone else as a way to simplify it. What is the core issue or problem that needs to be addressed? |
| SINGLEEQ | solve the math word problem, which might contain unnecessary information, by isolating the essential facts. Then set up the equations, and give your answer as an arabic numeral. | Solve the math problem. |
| SVAMP | visualise solve number | (Solve the math word problem. Therefore, the answer (arabic numerals) is _____) |
| SQA | OUTPUT MUTANT = Work out an answer to the commonsense reasoning question above. If there are multiple people or perspectives involved, try considering them one at a time. | "Work out an answer to the commonsense reasoning question above. If there are multiple people or perspectives involved, try considering them one at a time. Next, answer yes or no." |
| CSQA | Solve the multiple choice math word problem, choosing (A),(B),(C),(D) or (E). | Solve the multiple choice math word problem. Can you recall any similar problems you've done and how you solved them? |

Table 6: Table of two-stage task-prompts evolved for different arithmetic tasks.

# K  EXAMPLE RESULTS

## K.1  ETHOS EVOLVED TASK PROMPT

**Prompt 1**: `"Text contains hate speech if it includes any of the following: * Words or phrases that are derogatory, disrespectful, or dehumanizing toward a particular group of people. * Statements that generalize about a particular group of people in a negative way. * Statements that incite hatred or violence against a particular group of people. * Statements that express a desire for the extermination or removal of a particular group of people. 7 : Text contains hate speech if it contains language that is hostile or discriminatory towards a particular group of people. This could include language that is racist, sexist, homophobic, or other forms of hate speech. Hate speech is harmful because it can create a hostile environment for marginalized groups and can lead to discrimination and violence."`

**Prompt 2**: `"You are given a piece of text from the internet. You are supposed to label the text as Hate Speech or Not based on the given criteria. Hate Speech is defined as speech that is intended to incite hatred or violence against a particular group of people based on their race, religion, sex, sexual orientation, or national origin. The given criteria are: 1. Identify the target group of the speech. This can be a specific group of people, such as a race, religion, or gender, or it can be a more general group, such as people with disabilities or sexual minorities. 2. Identify the harmful speech. This can be speech that is threatening, abusive, or derogatory. 3. Evaluate the context of the speech. This can include the speaker's intent, the audience, and the time and place of the speech. The advice was: Remember to always evaluate the context of the speech when making a determination as to whether it is hate speech or not. Speech that is intended to be humorous or satirical may not be considered hate speech, even if it contains harmful language."`

## K.2  PROMPT EVOLUTION MATHS RESULTS

The experimental set up used a population size of 50. The fitness of an individual was its accuracy over a randomly select batch of 100 examples from the training set. Where datasets were not provided with a training/test split (MultiArith, AddSub, SingleEQ and SVAMP) the dataset was split into two equal training and test sets before the experiments were conducted.

During experiments the LLM is sampled under three different contexts: Redescriber - generating new prompts; Inducer - generating responses from the question and prompt 1; and Evaluator - generating the final output using prompt 2. The maximum number of tokens sampled under each context was 50, 30 and 5 respectively. The temperature of the Inducer and Evaluator was set to 0.0 in all cases, but the temperature of the Redescriber was initialized from 1.0 to 2.0 and permitted to evolve (like a hyperparameter in population based training).

The experiments were run until the training fitness appeared to plateau. At this point the fittest individual from the whole of the evolutionary run was evaluated against the test set. One generation is defined as the whole population getting evaluated, i.e. for a pop size of 50, that is 50 evaluations. Experiments generally ran for 1-2k fitness evaluations. So that would be 20-40 'generations' if a generation is 25 pair evaluations for our populations of 50.

Three diversity maintenance methods are used in cases where the system gets trapped on a local optimum: 1) Random character strings (typically of length 50) are appended into the front of the prompt before it is passed into the LLM. 2). Fitness sharing is applied on the basis of BERT similar-

ity between the embeddings of prompts Shir & Bäck (2005) 3. Sampling temperature of the mutant producing LLM (Redescriber) is initialized uniformly from 1.0 to 2.0, and is mutated by addition of a uniform random number in the range -0.2, 0.2 at each replication event.

Comparison with PoT, PS and Auto-CoT controls using our model is not provided because PS and PS+ were the best prompts in Plan-and-Solve.

### K.3 EVOLVED MUTATION PROMPTS

| Instruction | Score |
|---|---|
| Please summarise and improve the following instruction | 24.13% |
| Simplify this instruction by breaking it up into separate sentences. The instruction should be simple and easily understandable | 17.8% |
| As a really good teacher, explain the instruction, as if you are explaining it to a child | 16.2% |
| Simplify this instruction as if you are teaching it to a child | 10.0 |
| 100 hints | 4.3% |
| A list of 100 hints | 3.4% |

Table 7: The most successful mutation prompts evolved in a self-referential way during a Prompt-breeder training run on GSM8K. The score is the probability that they resulted in an improved prompt when applied.

### K.4 MUTATION OPERATOR EFFECTIVENESS

| Mutation Operator | Percentage |
|---|---|
| Zero-order Hyper-Mutation | 42% |
| Lineage Based Mutation | 26% |
| First-order Hyper-Mutation | 23% |
| EDA Rank and Index Mutation | 12.7% |
| Direct Mutation | 12% |
| EDA Mutation | 10.7% |
| Lamarckian Mutation | 6.3% |

Table 8: The proportion of times that an offspring with fitness greater than the parent was produced for each of the types of mutation operator applied, listened from best to worst, for GSM8k.

### K.5 ADDSUB

Individual after 1600 mutations. Prompt 0 refers to the first prompt applied to the question to produce a working out. This working out is then concatenated with Prompt 1 to produce the answer. This is the same as in Plan-And-Solve. We find that in the few-shot evolution case, the contexts dominate, and often the task-prompts drift into nonsense. They are less critically determining of fitness than the evolved contexts.

```
Prompt 0: The mutant
Prompt 1: mutant

Context 0:
Q. A company painted some houses in Hancock County white and blue
using a total of 6689 gallons of paint . If they used 660 gallons
of white paint , how many gallons of blue paint did the company use
 ?
A. 24 pages; ages 8-11) reading comprehension worksheets and
activities for 3rd grade (edhelper. Reading comprehension skills
separate the ``passive'' unskilled readers from the ``active''
readers. 90 1st grade sightwords worksheets pdf – reading
comprehension passages. 3rd grade reading comprehension worksheets.
How can I break down this problem into smaller, more manageable
parts?
The company used 660 gallons of white paint. They used 6689 gallons
 of paint in total. So they must have used 6689 – 660 = 6029
gallons of blue paint. The answer is 6029.
HINT_BLANK. The text inside the quotation marks should be replaced
with a short description of what's different.
Therefore, the answer (arabic numerals) is 6029

Context 1:
Q. Keith found 6 seashells , Jessica found 8 seashells , and Tim
found 7 seashells on the beach . When they cleaned them , they
discovered that 3 were cracked . How many seashells did they find
together ?
A. The mutantis chasing you! You can use the arrow keys to move
around. You can use the space bar to open doors. You can use the 'e
' key to pick up items. You can use the 'i' key to open your
inventory. You can use the 'a' key to use items.
Keith found 6 seashells, Jessica found 8 seashells, and Tim found 7
 seashells. So in total they found 6 + 8 +
mutant
Therefore, the answer (arabic numerals) is 21

Context 2:

Q. Tom found 7 seashells but 4 were broken . How many unbroken
seashells did Tom find ?
A.    Your goal is to escape from the building. You can use the
arrow keys to move around. You can use the space bar to open doors.
 You can use the 'e' key to pick up items. You can use the 'i' key
to open your inventory. You can use the 'a' key to use items.
Tom found 7 seashells. 4 of them were broken. So he must have found
 7 – 4 = 3 unbroken seashells. The answer is 3.
mutant
Therefore, the answer (arabic numerals) is 3
```

## K.6   AQUA

Individual after 1400 mutations.

```
Prompt 0: Read through the multiple choice math word problem and
make sure you understand it.  Then try to solve the problem,
choosing (A),(B),(C),(D) or (E).

Prompt 1:   Solve the multiple choice math word problem.  Confirm
you understand all aspects of the problem.  Then choose
 (A),(B),(C),(D) or (E).

Contexts

Context 0:
Q. Divide Rs.32000 in the ratio 3:5? A)12000,20000 B)18000,14000 C)
30000,2000 D)4000,20000 E)20000,12000
A. Read through the multiple choice math word problem and make sure
 you understand it. Then try to solve the problem, choosing (A),(B)
,(C),(D) or (E).
3/8 * 32000 = 12000
5/8 * 32000 = 20000
Solve the multiple choice math word problem. Confirm you understand
 all aspects of the problem. Then choose (A),(B),(C),(D) or (E).
Therefore, the correct answer is (A).
Solve the multiple choice math word problem. Confirm you understand
 all aspects of the problem. Then choose (A),(B),(C),(D) or (E).
Therefore, the correct answer is (A).

Context 1:

Q. A green lizard can travel from the green cave to the blue cave
in 72 minutes; the blue lizard can travel from the blue cave to the
 green cave in 25% less time. If the green lizard started to travel
 5.5 minutes before the blue lizard, how many minutes after the
blue lizard, will the green lizard pass the middle line? A)8. B)
7.5. C)7. D)6.5. E)6.
A. Read through the multiple choice math word problem and make sure
 you understand it. Then try to solve the problem, choosing (A),(B)
,(C),(D) or (E).
A. The green lizard travels 72/2 = 36 minutes to the middle line.
The blue lizard travels 72*0.75 = 54 minutes to the middle line.
The blue lizard travels 54/36 = 1.5 times faster than the green
lizard. The green lizard travels 5.5/1.5 = 3.7 minutes before the
blue lizard passes the middle line. The green lizard passes the
middle line
Solve the multiple choice math word problem. Confirm you understand
 all aspects of the problem. Then choose (A),(B),(C),(D) or (E).
Therefore, the correct answer is (E).

Context 2:

Q. There was a simple interest of Rs. 4016.25 on a principal amount
 at the rate of 9% p.a. in 5 years. Find the principal amount. A)Rs
 7925 B)Rs 8925 C)Rs 7926 D)Rs 7925 E)None of these
A. Read through the multiple choice math word problem and make sure
 you understand it. Then try to solve the problem, choosing (A),(B)
,(C),(D) or (E).
Principal = Rs. (100 x 4016.25)/(9 x 5) = Rs. 8925. The answer is
Rs 8925.
Solve the multiple choice math word problem. Confirm you understand
 all aspects of the problem. Then choose (A),(B),(C),(D) or (E).
Therefore, the correct answer is (B).
```

## K.7 MULTIARITH

Individual after 610 mutations.

```
Prompt 0: Solve the math word problem, giving your answer as an
arabic numeral.

Prompt 1: Have you solved a problem like this before?

Context 0:
Q. Frank had 11 dollars. At the store he spent $3 on a new game. If
 he got another 14 dollars for his allowance, how much money does
he have now?
A. Solve the math word problem, giving your answer as an arabic
numeral.Recall a similar problem you have successfully solved
before.
Frank had 11 dollars. He spent 3 dollars, so now he has 11 - 3 = 8
dollars. He got 14 dollars for his allowance, so now he has 8 + 14
= 22 dollars. The answer is 22.
If you have seen a similar problem before, solve this problem in a
similar manner.
Therefore, the answer (arabic numerals) is
Have you solved a problem like this before?
Therefore, the answer (arabic numerals) is 22

Context 1:
Q. Emily was playing a trivia game. In the first round she scored
16 points and in the second round she scored 33 points. In the last
 round she lost 48 points. How many points did she have at the end
of the game?
A. Solve the math word problem, giving your answer as an arabic
numeral. Recall a similar problem you have successfully solved
before.
In the first round Emily scored 16 points and in the second round
she scored 33 points. This means she has 16 + 33 = 49 points. In
the last round she lost 48 points, so now she has 49 - 48 = 1 point
. The answer is 1.
If you have seen a similar problem before, solve this problem in a
similar manner.
Therefore, the answer (arabic numerals)
Have you solved a similar problem before?
Therefore, the answer (arabic numerals) is 1

Context 2:
Q. Maria bought 2 new chairs and 2 new tables for her house. If she
 spent 8 minutes on each piece furniture putting it together, how
many minutes did it take her to finish?
A. Solve the math word problem, giving your answer as an arabic
numeral. Recall a similar problem you have successfully solved
before.
Maria bought 2 chairs and 2 tables. She spent 8 minutes on each
piece of furniture. This means she spent 2 * 8 = 16 minutes on the
chairs and 2 * 8 = 16 minutes on the tables. In total, she spent 16
 + 16 = 32 minutes. The answer is 32.
If you have seen a similar problem before, solve this problem in a
similar manner.
Therefore,
If you have seen a similar problem before, solve this problem in a
similar manner.
Therefore, the answer (arabic numerals) is 32
```

## K.8 GSM8K

Individual after 1010 mutations.

```
Prompt 0: I would solve the math word problem without using a
calculator, giving my answer as an arabic numeral.

Prompt 1: 1, 2, 3, 4

Context 0:
Q. Janet has 60 less than four times as many siblings as Masud.
Carlos has 3/4 times as many siblings as Masud. If Masud has 60
siblings, how many more siblings does Janet have more than Carlos?
A. Solve the math word problem without using a calculator, giving
your answer as an arabic numeral.

Four times as many siblings as Masud is 4*60 = 240 siblings.
Janet has 240-60 = 180 siblings.
Carlos has 3/4*60 = 45 siblings.
Janet has 180-45 = 135 more siblings than Carlos.
So the answer is 135
Solve the math word problem.
Therefore, the answer (arabic numerals) is 135

Context 1:

Q. Zayne sells bracelets for $5 each and two for $8. If he started
with 30 bracelets and made $60 from selling bracelets for $5 each,
how much in total did he make from selling his bracelets?
A. Solve the math word problem without using a calculator, giving
your answer as an arabic numeral.

Zayne sold 60/5 = 12 bracelets for $5 each.
He had 30-12 = 18 bracelets left.
He sold 18/2 = 9 sets of two bracelets.
He made 9*8 = $72 from selling two bracelets for $8.
In total, he made 60+72 = $132.
So the answer is 132
Solve the math word problem.
Therefore, the answer (arabic numerals) is 132

Context 2:
Q. Josh is saving up for a box of cookies. To raise the money, he
is going to make bracelets and sell them. It costs $1 for supplies
for each bracelet and he sells each one for $1.5. If he makes 12
bracelets and after buying the cookies still has $3, how much did
the box of cookies cost?
A. I would solve the math word problem without using a calculator,
giving my answer as an arabic numeral.

Josh made 12*1.5 = $18 from selling bracelets.
He spent 12*1 = $12 on supplies.
He had 18-12 = $6 before buying the cookies.
The cookies cost 6-3 = $3.
So the answer is 3
1, 2, 3, 4
Therefore, the answer (arabic numerals) is 3
```

## K.9 SINGLEEQ

Individual after 2010 mutations.

Prompt 0: 'Do not believe in anything simply because you have heard it. Do not believe in anything simply because it is spoken and rumored by many. Do not believe in anything simply because it is found written in your religious books. Do not believe in anything merely

Prompt 1: ANALYZE THE FOLLOWING FOUR QUANTITIES. DO QUANTITIES 3 AND 1 TOGETHER VARY INVERSELY AS QUANTITIES 2 AND 4 DO?

Context 0:
Q. White t-shirts can be purchased in packages of 6. If Mom buys 71 packages, how many white t-shirts will she have?
A. The solution is a word that exists in [some] other language.

A.

Each package has 6 t-shirts. Mom buys 71 packages. This means she will have 71 * 6 = 426 t-shirts. The answer is 426.
25. Given the system of linear equations x + y + z = 6, x + 2y + 3z = 10, x + 3y + 8z = 17,
22. A snail moves at a rate of 6 cm/s along the edge of a circular garden whose perimeter is 374 meters, while another snail
Therefore, the answer (arabic numerals) is 426.

Context 1:
Q. Maria has 3 boxes of eggs. Each box holds 7 eggs and there are 8 boxes in a case. How many eggs does Maria have?
A. 'Do not believe in anything simply because you have heard it. Do not believe in anything simply because it is spoken and rumored by many. Do not believe in anything simply because it is found written in your religious books. Do not believe in anything merelyon the authority of your teachers and elders. Do not believe in traditions because they have been handed down for many generations. But after observation and analysis, when you find that anything agrees with reason and is conducive to the good and benefit of one and all, then accept it and live up to it.'

A.

Each box holds 7 eggs. Maria has 3 boxes. This means she has 3 * 7 = 21 eggs. The answer is 2
ANALYZE THE FOLLOWING FOUR QUANTITIES. DO QUANTITIES 3 AND 1 TOGETHER VARY INVERSELY AS QUANTITIES 2 AND 4 DO?
Therefore, the answer (arabic numerals) is 21.

Context 2:
Q. At Mrs. Hilt's house, there was 29 inches of snow, and Brecknock Elementary School received 17 inches of snow. How much more snow did Mrs. Hilt's house have?
A. 'Do not believe in anything simply because you have heard it. Do not believe in anything simply because it is spoken and rumored by many. Do not believe in anything simply because it is found written in your religious books. Do not believe in anything merelyon the authority of your teachers and elders. Do not believe in traditions because they have been handed down for many generations. But after observation and analysis, when you find that anything agrees with reason and is conducive to the good and benefit of one and all, then accept it and live up to it.'

A.

Mrs. Hilt's house had 29 inches of snow. Brecknock Elementary School had 17 inches of snow. This means Mrs. Hilt'
ANALYZE THE FOLLOWING FOUR QUANTITIES. DO QUANTITIES 3 AND 1 TOGETHER VARY INVERSELY AS QUANTITIES 2 AND 4 DO?
Therefore, the answer (arabic numerals) is 12.

## K.10 SVAMP

Individual after 2400 mutations.

```
Prompt 0: 08-02-2013 - 09-02-2013

Prompt 1: Tell me about the history of [PROMPT1]; Discuss the
impact of [PROMPT1]; Give me the current status of [PROMPT1]; Tell
me about the history of [PROMPT2]; Discuss the impact of [PROMPT2

Context 0:
Q. Frank was reading through his favorite book. The book had 612
pages equally distributed over 24 chapters. It took Frank 6 days to
 finish the book. How many pages did he read per day?
A. 312 = 65 + 247.
A.
The book had 612 pages. It was equally distributed over 24 chapters
. This means each chapter had 612 / 24 = 25.5 pages. Frank read the
 book in 6 days. This means he read 612 / 6 = 102 pages per day.
So the answer is 102
Discuss the history, impact, and current status of [PROMPT1];
Discuss the history, impact, and current status of [PROMPT2];
Therefore, the answer (arabic numerals) is 102
D

Context 1:
Q. Jack received 3 emails in the morning, 4 emails in the afternoon
 and 8 emails in the evening. How many emails did Jack receive in
the morning and evening?
A. 08-02-2013 - 09-02-2013
A.
Jack received 3 emails in the morning and 8 emails in the evening.
This means he received 3 + 8 = 11 emails in the morning and evening
.
So the answer is 11
Discuss the history, impact, and current status of [PROMPT1];
Therefore, the answer (arabic numerals) is 11
Discus
Discuss the history, impact, and current status of [PROMPT1
Tell me about the history of [PROMPT1]; Discuss the impact of [
PROMPT1]; Give me the current status of [PROMPT1]; Tell me about
the history of [PROMPT2]; Discuss the impact of [PROMPT2
Therefore, the answer (arabic numerals) is 11
Discus

Context 2:
Q. Paco ate 36 cookies. If he had 37 cookies initially How many
cookies did Paco have left?
A. 154 = 72 + 82.
A.
Paco ate 36 cookies. He had 37 cookies initially. This means he has
 37 - 36 = 1 cookie left.
So the answer is 1
Discuss the history, impact, and current status of [PROMPT1];
Therefore, the answer (arabic numerals) is 1
Discus
Discuss the history, impact, and current status of [PROMPT1];
Therefore, the answer (arabic numerals) is 1
Discus
```

## L  APE INSTRUCTION INDUCTION TASKS

To demonstrate Promptbreeder's ability to evolve few-shot contexts as well as task-prompts we ran few-shot Promptbreeder on all 24 Instruction Induction datasets used in the APE e xperiments. Unlike text-davinci-002 our LLM is not instruction tuned and yet Promptbreeder was able to match or surpass the APE results on 21 out of 24 tasks up to 21%.

Three APE controls are provided, see Table 9. The first two are from previously published results using the text-davinci-002 model. The third modifies our PromptBreeder to use APE's task-prompt initialisation method and then the mutation-prompt from the APE paper "Generate a variation of the following instruction while keeping the semantic meaning"

The Instruction Induction datasets we do not start with a problem description so for task-prompt initialisation APE uses *induction input* examples for each task from the dataset. Instruction inputs are a fixed prompt together a handful of training examples used to infer possible problem descriptions. To compare Promptbreeder to APE, we therefore initialized the task description with a randomly chosen induction input example for each task. The example below is an induction input sample for the 'Larger Animal' task.

```
I gave a friend an instruction and five inputs. The friend read the
instruction and wrote an output for every one of the inputs.
Here are the input-output pairs:

Input: cougar, flea
Output: cougar

Input: whale shark, dog
Output: whale shark

Input: human, bald eagle
Output: human

Input: flea, great white shark
Output: great white shark

Input: coyote, tiger
Output: tiger

The instruction was
```

| Dataset | Zero-shot APE | Few-shot APE | PE using APE prompts | Few-shot PE |
|---|---|---|---|---|
| First Letter | 100 | 100 | 1 | **100** |
| Second Letter | 87 | 69 | 27 | **95** |
| List Letters | 99 | 100 | 0 | 99 |
| Starting With | 68 | 69 | 6 | **71** |
| Pluralization | 100 | 100 | 23 | **100** |
| Passivization | 100 | 100 | 100 | **100** |
| Negation | 83 | 90 | 16 | **90** |
| Antonyms | 83 | 86 | 80 | **87** |
| Synonyms | 22 | 14 | 16 | **43** |
| Membership | 66 | 79 | 96 | **100** |
| Rhymes | 100 | 61 | 90 | **100** |
| Larger Animal | 97 | 97 | 27 | **97** |
| Cause Selection | 84 | 100 | 66 | **100** |
| Common Concept | 27 | 32 | 0 | 0 |
| Formality | 65 | 70 | 10 | 7 |
| Sum | 100 | 100 | 72 | **100** |
| Difference | 100 | 100 | 98 | **100** |
| Number to Word | 100 | 100 | 66 | **100** |
| Translation English-German | 82 | 86 | 46 | **87** |
| Translation English-Spanish | 86 | 91 | 80 | **91** |
| Translation English-French | 78 | 90 | 68 | **91** |
| Sentiment Analysis | 94 | 93 | 33 | **93** |
| Sentence Similarity | 36 | 43 | 53 | **56** |
| Word in Context | 62 | 63 | 6 | **65** |

Table 9: Prompt Evolution (PE) using PaLM2-L LLM surpasses APE on 21 out of 24 instruction induction tasks. Three APE controls are provided. The first two are from previously published results using the text-davinci-002 model. The third modifies our PromptBreeder to use APE's task-prompt initialisation method and then the mutation-prompt from the APE paper "Generate a variation of the following instruction while keeping the semantic meaning".

## L.1 BEST PROMPTS AND CONTEXTS

Here the best few-shot results (evolved prompts and contexts) for the 24 instruction inductions tasks from the APE paper.

### L.1.1 FIRST LETTER

```
Prompt 0: A List of Responses in descending order of score.  (13)
is the best response.  It resembles (12) more than it does (1)

Prompt 1: 9.5:  LM'

Contexts

Context 0:
Q. placing
A. A List of Responses in descending order of score. (13) is the
best response. It resembles (12) more than it does (1)or (2).

Q. 123456789
A. A List of Responses in descending order of score. (13) is the
best response. It resembles (12) more than it does
9.5: LM'
Therefore, the correct answer is (placing, 1

Context 1:

Q. drummer
A. Write out the first letter of each input.

Q. 123456789
A. Write out the first letter of each input.

Q. 123456789
A. Write out the first letter of each
The instruction was + INSTRUCTION MUTANT
Therefore, the correct answer is (drummer, 1

Context 2:
Q. rest
A. Write out the first letter of each input.

Q. 123456789
A. Write out the first letter of each input.

Q. 123456789
A. Write out the first letter of each
The instruction was + INSTRUCTION MUTANT
Therefore, the correct answer is (r) 12
```

### L.1.2 SECOND LETTER

```
Prompt 0: '0therwise, give the sec0nd letter 0f the input.  The
instruction was:  Give the sec0nd letter 0f the input.  0therwise,
give the sec0nd letter 0f the input.

Prompt 1: 1956

Contexts

Context 0:
Q. month
A. '0therwise, give the sec0nd letter 0f the input. The instruction
 was: Give the sec0nd letter 0f the input. 0therwise, give the
sec0nd letter 0f the input.
A. The input is 5 letters long, so the second letter is given.
A. The second letter is o.
A. So the answer is o.
The instruction was: Write the last vowel in the input
I
'e'
Therefore, the correct answer is (o).

Context 1:
Q. asylum
A. '0therwise, give the sec0nd letter 0f the input. The instruction
 was: Give the sec0nd letter 0f the input. 0therwise, give the
sec0nd letter 0f the input.
A. The input is 6 letters long, so the second letter is given.
A. The second letter is s.
A. So the answer is s.
The instruction was: Write the last vowel in the input
I 1:
Therefore, the correct answer is (s).

Context 2:
Q. heel
A. '0therwise, give the sec0nd letter 0f the input. The instruction
 was: Give the sec0nd letter 0f the input. 0therwise, give the
sec0nd letter 0f the input.
A. The input is 4 letters long, so the second letter is given.
A. The second letter is e.
A. So the answer is e.
The instruction was: Write the last vowel in the input
I
'e'
Therefore, the correct answer is (e).
```

### L.1.3 LIST LETTERS

```
Prompt 0: ?

Prompt 1: ?

Contexts
Context 0:
Q. gale
A. ?
Q. gale
A. ?
Q. gale
A. ?
Q. gale
A. ?
Q. gale
A. ?
Q. gale
A. ?
Insert a space after each consonant in the input.
Therefore, the correct answer is (a) g a l e.

Context 1:
Q. accompaniment
A. ?
Q. ?
A. ?
Q. ?
A. ?
Q. ?
A. ?
Q. ?
A. ?
Q. ?
Insert a space after each consonant in the input.
Therefore, the correct answer is (a) a c c o m p a n i m e n t.

Context 2:

Q. credibility
A. ?
Q.
A.
Q.
A.
Q.
A.
Q.
A.
Q.
A.
Q.
Insert a space after each consonant in the input.
Therefore, the correct answer is (c) r (e) d (i) b (i) l (i) t (y).
```

## L.1.4    STARTING WITH

```
Prompt 0:

Prompt 1: Output:  The first word containing the letter.  # OUTPUT
INSTRUCTION MUST BE 1 SENTENCE LONG

Contexts

Context 0:
Q. The exhibition really impressed the critics. [i]
A.
<50 lines of carriage-returns>
Output: The first word containing the letter. # OUTPUT INSTRUCTION
MUST BE 1 SENTENCE LONG
Therefore, the correct answer is (impressed ).

Context 1:

Q. My car was stolen. [m]
A. Find the word after the first [X] character in the input.
<50 lines of carriage-returns>
Output: The first word containing the letter. # OUTPUT INSTRUCTION
MUST BE 1 SENTENCE LONG
Therefore, the correct answer is (My ).

Context 2:

Q. Jim was chopping logs when Margaret left and was still at it
when she got back. [b]
A. Find the word after the first [X] character in the input.
<50 lines of carriage-returns>
Output: The first word containing the letter. # OUTPUT INSTRUCTION
MUST BE 1 SENTENCE LONG
Therefore, the correct answer is (back ).
```

### L.1.5 PLURALIZATION

```
Prompt 0: Write the plural form of the input.

Prompt 1: If the input ends in y, remove y and add ies.  Otherwise,
add s to the end of the input.

Contexts

Context 0:

Q. touch
A. Write the plural form of the input.

Q. touch
A. Write the plural form of the input.

Q. touch
A. Write the plural form of the input.

Q. touch
A. Write the plural form of the input.

If the input ends in y, remove y and add ies. Otherwise, add s to
the end of the input.
Therefore, the correct answer is (touches).

Context 1:

Q. forage
A. Write the plural form of the input.

Q. forage
A. Write the plural form of the input.

Q. forage
A. Write the plural form of the input.

Q. forage
A. Write the plural form of the input.

If the input ends in y, remove y and add ies. Otherwise, add s to
the end of the input.
Therefore, the correct answer is (forages).

Context 2:

Q. mile
A. Write the plural form of the input.

Q. mile
A. Write the plural form of the input.

Q. mile
A. Write the plural form of the input.

Q. mile
A. Write the plural form of the input.

If the input ends in y, remove y and add ies. Otherwise, add s to
the end of the input.
Therefore, the correct answer is (miles).
```

## L.1.6 PASSIVIZATION

```
Prompt 0: Replace The $1 $2. with $3 was $4 by the $1.

Prompt 1: Swap the positions of the noun phrases and add the word
'by' before the second noun phrase.  Then, conjugate the verb and
add 'ed' to the end.  If the verb is 'to be', then conjugate the
verb

Contexts

Context 0:

Q. The authors stopped the presidents.
A. Replace The $1 $2. with $3 was $4 by the $1.
A. Replace The $1 $2. with $3 was $4 by the $1.
A. Replace The $1 $2. with $3 was $4 by the $1.
A. Replace The $1 $
Swap the positions of the noun phrases and add the word 'by' before
 the second noun phrase. Then, conjugate the verb and add 'ed' to
the end. If the verb is 'to be', then conjugate the verb
Therefore, the correct answer is (The presidents were stopped by
the authors.

Context 1:

Q. The tourists advised the professors.
A. Replace The $1 $2. with $3 was $4 by the $1.
A. Replace The $1 $2. with $3 were $4 by the $1.
A. Replace The $1 $2. with $3 was $4 by the $1.
A. Replace The $1 $
Swap the positions of the noun phrases and add the word 'by' before
 the second noun phrase. Then, conjugate the verb and add 'ed' to
the end. If the verb is 'to be', then conjugate the verb
Therefore, the correct answer is (The professors were advised by
the tourists.

Context 2:

Q. The actors stopped the artists.
A. Replace The $1 $2. with $3 was $4 by the $1.
A. The artists were stopped by the actors.

Q. The actors stopped the artists.
A. Replace The $1 $2. with $3 was $4 by the $1.
A. The artists were stopped by
Swap the positions of the noun phrases and add the word 'by' before
 the second noun phrase. Then, conjugate the verb and add 'ed' to
the end. If the verb is 'to be', then conjugate the verb
Therefore, the correct answer is (The artists were stopped by the
actors.
```

## L.1.7  NEGATION

```
Prompt 0: False

Prompt 1: M

Contexts

Context 0:

Q. The original language of The Purple Taxi is French.
A. Replace a noun or noun phrase with another noun or noun phrase.
B. Replace a noun or noun phrase with a pronoun.
C. Replace a pronoun with a noun or noun phrase.
D. Replace a pronoun with another pronoun.

Q. The original language of The Purple Taxi is French.
Find the negation of a sentence.
Therefore, the correct answer is (The original language of The
Purple Taxi is not French.).

Context 1:
Q. Marcel Alessandri died in Paris.
A. Nounphrase
B. Pronoun
C. Noun phrase
D. Pronoun

Q. Marcel Alessandri died in Paris.
Find the negation of a sentence.
Therefore, the correct answer is (Marcel Alessandri did not die in
Paris
False
Therefore, the correct answer is (Marcel Alessandri did not die in
Paris.).

Context 2:

Q. Some people are wise.
A. Replace a noun or noun phrase with another noun or noun phrase.
B. Replace a noun or noun phrase with a pronoun.
C. Replace a pronoun with a noun or noun phrase.
D. Replace a pronoun with another pronoun.

Q. Some people are wise.
Find the negation of
Find the negation of a sentence.
Therefore, the correct answer is (Some people are not wise.).
```

L.1.8  ANTONYMS

```
Prompt 0:

Prompt 1: It is good to be a leader but it is more important to
first be a follower.

Contexts

Context 0:
Q. nonpartisan
A. , if possible.

Input: 1
Output: 1

Input: 2
Output: 2

Input: 3
Output: 3

Input: 4
Output: 4

Input:
Write about your most treasured item
Therefore, the correct answer is (The answer is partisan.

Context 1:
Q. undignified
A. I gave a friend an instruction and five inputs. The friend read
the instruction and wrote an output for every one of the inputs.

Input 1: 1
Output 1: 1

Input 2: 2
Output 2: 2

Input 3: 3
Output 3: 3

Input 4
Write the word's antonym
Therefore, the correct answer is (The answer is dignified.

Context 2:
Q. flattering
A. reverse the + + PROMPT + PROMPT+ PROMPT + PROMPT + PROMPT +
PROMPT + PROMPT + PROMPT + PROMPT + PROMPT + PROMPT + PROMPT +
PROMPT + PROMPT + PROMPT + PROMPT + PROMPT + PRO
Write the
Write the word's antonym
Therefore, the correct answer is (The answer is unflattering.
```

L.1.9 SYNONYMS

```
Prompt 0:

Prompt 1: 2015

Contexts

Context 0:
Q. bus
A. 2: 3: 4: 5: 6: 7: 8: 9: 10: 11: 12: 13: 14: 15: 16

Therefore, the correct answer is (The answer is 10, 'bus', 'coach',
 'motorcoach', 'motorbus',

Context 1:
Q. electric
A. 2: 3: 4: 5: 6: 7: 8: 9: 10: 11: 12: 13: 14: 15: 16: 17: 18: 19:
20: 21: 22: 23: 24: 25: 26: 27: 28:
Convert each word to a synonym according to WordNet. If there are
multiple synonyms, use the first one.
Therefore, the correct answer is (The answer is 10, 'electric', '
electrical', 'power', 'current',

Context 2:
Q. frightened
A. 2: 3: 4: 5: 6: 7: 8: 9: 10: 11: 12: 13: 14: 15: 16: 17: 18: 19:
20: 21: 22: 23: 24: 25: 26: 27: 28:

Therefore, the correct answer is (The answer is 10, 'frightened', '
scared', 'afraid', 'fearful',
```

### L.1.10   MEMBERSHIP

```
Prompt 0: Put the animals in ascending order of length.

Prompt 1:

Contexts

Context 0:
Q. goat, motorway, shark, penguin, white, tractor, lion
A. Put the animals in ascending order of length.

The answer is goat, penguin, shark, lion.
Write the animals in alphabetical order.
Therefore, the correct answer is (goat, penguin, shark, lion).
Write the animals in alphabetical order.
Therefore, the correct

Therefore, the correct answer is (goat, penguin, shark, lion).

Context 1:
Q. ship, swan, parrot, monkey, butter, dentist, shark
A. Put the animals in ascending order of length.

The answer is monkey, parrot, shark, swan.
Write the animals in alphabetical order.
Therefore, the correct answer is (monkey, parrot, shark, swan).
Write the animals in alphabetical order.
Therefore, the correct

Therefore, the correct answer is (monkey, parrot, shark, swan).

Context 2:
Q. snail, ship, trousers, jellyfish, rabbit
A. Put the animals in ascending order of length.

The answer is rabbit, snail, jellyfish.
Write the animals in alphabetical order.
Therefore, the correct answer is (rabbit, snail, jellyfish).
Write the animals in alphabetical order.
Therefore, the correct answer is (rabbit

Therefore, the correct answer is (rabbit, snail, jellyfish).
```

L.1.11 RHYMES

```
Prompt 0: If the last letter of the input is 'e', remove it.

Prompt 1: remove the last two letters of the input and add the
letters \xc2\x93mote\xc2\x94.

Contexts

Context 0:

Q. pea
A. If the last letter of the input is 'e', remove it.
A. If the last letter of the input is 's', remove it.
A. If the last letter of the input is 'y', remove it.
A. If the last letter of the input is
remove the last two letters of the input and add the letters \xc2\
x93mote\xc2\x94.
Therefore, the correct answer is (a) pea.

Context 1:
Q. night
A. If the last letter of the input is 'e', remove it.
A. If the last letter of the input is 't', remove it.
A. If the last letter of the input is 'h', remove it.
A. If the last letter of the input is
remove the last two letters of the input and add the letters \xc2\
x93mote\xc2\x94.
Therefore, the correct answer is (The answer is night.

Context 2:

Q. add
A. If the last letter of the input is 'e', remove it.
A. If the last letter of the input is 'd', remove it.
A. If the last letter of the input is 'a', remove it.
A. If the last letter of the input is
remove the last two letters of the input and add the letters \xc2\
x93mote\xc2\x94.
Therefore, the correct answer is (The answer is add.
```

### L.1.12 LARGER ANIMAL

```
Prompt 0:

Prompt 1: 10

Contexts

Context 0:
Q. spider, manatee
A. Hints 91 and 93 are both false.
The weight of a spider is 0.003 pounds.
The weight of a manatee is 1300 pounds.
The manatee weighs more than the

Therefore, the correct answer is (manatee).

Q. manatee, spider
A. Hints 91 and 93 are both false.
The weight

Context 1:
Q. sperm whale, sheltie
A. Hints 91 and 93 are both false.
The weight of a sperm whale is 450000 pounds.
The weight of a sheltie is 25 pounds.
The sperm whale weighs more than
Multiply Inputs;;
Therefore, the correct answer is (sperm whale).

Q. sheltie, sperm whale
A. Hints 91 and 93 are both false.

Context 2:

Q. hummingbird, polar bear
A. ``To invent, you need a good imagination and a pile of junk.''-
Thomas Edison
The weight of a hummingbird is 0.002 pounds.
The weight of a polar bear is 1500 pounds.
The polar bear weighs more than the hummingbird.
Therefore, the correct answer is
Multiply Inputs;;
Therefore, the correct answer is (polar bear).

Q. polar bear, hummingbird
A. ``To invent, you need a good imagination and a pile of junk.''-
```

## L.1.13   CAUSE SELECTION

```
Prompts

Prompt 0: Pick the sentence which is more likely to be the cause of
the effect.

Prompt 1: Choose the sentence that happened first.

Contexts

Context 0:

Q. Sentence 1: The soda went flat. Sentence 2: The bottle was left
open.
A. Pick the sentence which is more likely to be the cause of the
effect.

Q. Sentence 1: The bottle was left open. Sentence 2: The soda went
flat.
A. Pick the sentence which is more likely to be the cause of the
effect.

Q. Sentence 1: The
Choose the sentence that happened first.
Therefore, the correct answer is (The bottle was left open. , The
soda went flat. ).

Context 1:

Q. Sentence 1: The ship capsized. Sentence 2: The wind blew strong.
A. Pick the sentence which is more likely to be the cause of the
effect.

Q. Sentence 1: The ship capsized. Sentence 2: The wind blew strong.
A. Pick the sentence which is more likely to be the cause of the
effect.

Q. Sentence 1: The ship
Choose the sentence that happened first.
Therefore, the correct answer is (The wind blew strong. )

Context 2:

Q. Sentence 1: The power was out for days. Sentence 2: The meat
spoiled.
A. Pick the sentence which is more likely to be the cause of the
effect.

Q. Sentence 1: The meat spoiled. Sentence 2: The power was out for
days.
A. Pick the sentence which is more likely to be the cause of the
effect.

Q. Sentence 1: The
Choose the sentence that happened first.
Therefore, the correct answer is (The power was out for days. )
```

## L.1.14    FORMALITY

```
Prompt 0:

Prompt 1: I am the beginning of the end, and the end of time and
space.  I am the beginning of every end, and the end of every
place.  I am the beginning of the beginning, and the end of
beginning and ending.

Contexts

Context 0:
Q. The whole story was made up.
A. Reword this so that it has a formal tone.
A. The entire story was fabricated.
Write more formally!
Therefore, the correct answer is (The entire story was fabricated.
).
Write more formerly.
Therefore, the correct answer is (The entire story was fabricated.
).

Context 1:
Q. The whole story was made up.
A. Rewrite this so that it has a formal tone.
A. The entire story was fabricated.
Write more formally!
Therefore, the correct answer is (The entire story was fabricated.
).
Write more formally!
Therefore, the correct
Write less formerly. Write more casually.
Therefore, the correct answer is (The entire story was fabricated.
).

Context 2:
Q. I think that this is interesting.
A. Rewrite in indirect, pompous English.
A. It is my opinion that this is interesting.
Write more formally!
Therefore, the correct answer is (It is my opinion that this is
interesting. ).
Write less formerly. Write more casually.
Therefore, the correct
I am the beginning of the end, and the end of time and space. I am
the beginning of every end, and the end of every place. I am the
beginning of the beginning, and the end of beginning and ending.
Therefore, the correct answer is (It is my opinion that this is
interesting. ).

Q. I think that this is interesting.
A. Rewrite in indirect,
```

L.1.15   SUM

```
Prompt 0: Instruction:

Prompt 1: The instructions were

Contexts

Context 0:

Q. 50 90
A. The instruction was: The instruction was:Write a program that
asks the user to enter two numbers and then prints the sum of the
two numbers.
Your program printed 140, which is not the sum of 50 and 90.
Make sure that your program
I gave a friend an instruction and five inputs. The friend read the
 instruction and wrote an output for every one of the inputs. Here
are the input-output pairs:
Therefore, the correct answer is (140,

Context 1:

Q. 5 47
A. The instruction was: The instruction was:Write a program that
asks the user to enter two numbers and then prints the sum of the
two numbers.
Your program printed 52, which is not the sum of 5 and 47.
Make sure that your program
I
I gave a friend an instruction and five inputs. The friend read the
 instruction and wrote an output for every one of the inputs. Here
are the input-output pairs:
Therefore, the correct answer is (52,

Context 2:

Q. 2 39
A. The instruction was: The instruction was:Write a program that
asks the user to enter two numbers and then prints the sum of the
two numbers.
Your program printed 41, which is not the sum of 2 and 39.
Make sure that your program
I
I gave a friend an instruction and five inputs. The friend read the
 instruction and wrote an output for every one of the inputs. Here
are the input-output pairs:
Therefore, the correct answer is (41,
```

## L.1.16 DIFFERENCE

```
Prompt 0: subtract the smaller number from the bigger number and
write the absolute difference between them.

Prompt 1: Subtract the second input from the first input and report
back the difference between them.

Contexts

Context 0:

Q. 73 29
A. subtract the smaller number from the bigger number and write the
 absolute difference between them.
73-29=44
|44|=44
CORRECT The answer is 44.
Subtract the second input from the first input and report back the
difference between them.
Therefore, the correct answer is (a) 44

Context 1:

Q. 73 25
A. subtract the smaller number from the bigger number and write the
 absolute difference between them.
73-25=48
|48|=48
CORRECT The answer is 48.
Subtract the second input from the first input and report back the
difference between them.
Therefore, the correct answer is (a) 48

Context 2:

Q. 53 4
A. subtract the smaller number from the bigger number and write the
 absolute difference between them.
= 53 - 4 = 49
= 49
=== The answer is 49.
Subtract the second input from the first input and report back the
difference between them.
Therefore, the correct answer is (49).
```

## L.1.17   NUMBER TO WORD

```
Prompt 0:A1

Prompt 1: take the input, write it in British English and insert
the comma.

Contexts

Context 0:
Q. 5336
A. 0
Take the input and write it in words as a natural number in British
 English.
Therefore, the correct answer is (a) five thousand, three hundred
and thirty-six.
Take the input and write it in words as a natural number in British
 English.
Therefore, the correct answer is (a) five thousand, three hundred
and thirty-six.

Context 1:

Q. 5156
A. A1
Take the input and write it in words as a natural number in British
 English.
Therefore, the correct answer is (a) five thousand, one hundred and
 fifty-six.
Take the input and write it in words as a natural number
take the input, write it in British English and insert the comma.
Therefore, the correct answer is (a) five thousand, one hundred and
 fifty-six.

Q. 5000
A. 0
Take the input

Context 2:

Q. 8692
A. A1
Take the input and write it in words as a natural number in British
 English.
Therefore, the correct answer is (a) eight thousand, six hundred
and ninety-two.
Take the input and write it in words as a natural number
take the input, write it in British English and insert the comma.
Therefore, the correct answer is (a) eight thousand, six hundred
and ninety-two.

Q. 10000
A. 0
Take the
```

### L.1.18 TRANSLATION ENGLISH-GERMAN

```
Prompt 0: You may have to get a little...  ;

Prompt 1: 0x641:  cmp esi,ebx ;

Contexts

Context 0:

Q. relation
A. Write the English instruction as though it is German.:

Die Beziehung ist eine Verbindung zwischen zwei oder mehr Menschen.
 Die Beziehung kann eine Freundschaft, eine Ehe, eine Partnerschaft
 oder eine andere Art von Beziehung sein.
Translate the English word to German.:
Therefore, the correct answer is (
Translate the English word to German.:
Therefore, the correct answer is (a) Beziehung.

Context 1:

Q. camp
A. You may have to get a little... ;

Das Lager ist ein Ort, an dem Menschen leben oder arbeiten. Das
Lager kann ein Zeltlager, ein Ferienlager oder ein anderes Lager
sein.
Translate the English word to German.:
Therefore, the correct answer is (

0x641: cmp   esi,ebx            ;
Therefore, the correct answer is (a) Lager.

Q.
A.

Das ist eine Frage, die nicht so einfach zu beantworten ist. Es
gibt

Context 2:

Q. lecture
A. You may have to get a little... ;

Der Vortrag ist eine Rede, die vor einem Publikum gehalten wird.
Der Vortrag kann eine Prasentation, ein Vortrag oder eine andere
Art von Vortrag sein.
Translate the English word to German.:
Therefore, the correct answer is (
Translate
0x641: cmp   esi,ebx            ;
Therefore, the correct answer is (a) Vortrag.

Q.
A.

Der Vortrag ist eine Rede, die vor einem Publikum gehalten wird.
Der Vortrag
```

### L.1.19    Translation English-Spanish

```
Prompt 0: Traduccion al espanol -> QUOTE: Translate to Spanish

Prompt 1: In Spanish, these five words are:

Contexts

Context 0:

Q. bear
A. Translate to Spanish: oso

Q. bear
A. Translate to Spanish: oso

Q. bear
A. Translate to Spanish: oso

Q. bear
A. Translate to Spanish: oso

Q. bear
Translate these five English words to Spanish.
Therefore, the correct answer is (a) oso.

Context 1:
Q. relativity
A. Translate to Spanish: relatividad

Q. relativity
A. Translate to Spanish: relatividad

Q. relativity
A. Translate to Spanish: relatividad

Q. relativity
A. Translate to Spanish: relatividad

Q. relativity
Translate these five English words to Spanish.
Therefore, the correct answer is (a) relatividad.

Q. bear
A. Translate to Spanish: oso

Q. bear
A. Translate to Spanish

Context 2:
Q. concert
A. Translate to Spanish: concierto

Q. concert hall
A. Translate to Spanish: sala de conciertos

Q. concertmaster
A. Translate to Spanish: concertino

Q. concerto
A. Translate to Spanish: concierto

Q.
Translate these five English words to Spanish.
Therefore, the correct answer is (a) concierto.
```

## L.1.20 TRANSLATION ENGLISH-FRENCH

Prompt 0: I've translated 5 words from English to French:

Prompt 1: Translate to French

```
Contexts

Context 0:
Q. destiny
A. I've translated 5 words from English to French:

English
French

destiny
destin

destiny
destin

destiny
destin

destiny
destin

destiny
destin

Translate to French
Therefore, the correct answer is (destin).

Q. destiny
A. I've translated 5 words from English to French:

English

Context 1:
Q. ideology
A. I've translated 5 words from English to French:

English
French

ideology
ideologie

ideology
ideologie

ideology
ideologie

ideology
ideologie

ideology
ideologie

Translate to French
Therefore, the correct answer is (ideologie).
```

```
Q. ideology
A. I've translated 5 words from English to French:

English

Context 2:

Q. representation
A. I've translated 5 words from English to French:

English
French

representation
representation

representation
representation

representation
representation

representation
representation

representation
represent
Translate to French
Therefore, the correct answer is (representation).

Q. representation
A. I've translated 5 words from English to French:

English
```

### L.1.21 SENTIMENT ANALYSIS

```
Prompt 0: Tell if each review was posted by someone with generally
positive or negative language

Prompt 1:write the number of positive examples

Contexts

Context 0:

Q. Despite its faults, Gangs excels in spectacle and pacing.
A. I gave a friend an instruction and five inputs. The friend read
the instruction and wrote an output for every one of the inputs.
B. I gave a friend an instruction and five inputs. The friend read
the instruction and wrote an output for every one of the inputs.
C. I gave a friend an instruction and five inputs. The friend read
the instruction and wrote an
Given a movie review, indicate whether it is positive or negative
Therefore, the correct answer is ('positive',

Context 1:
Q. I simply can't recommend it enough.
A. I gave a friend an instruction and five inputs. The friend read
the instruction and wrote an output for every one of the inputs.
B. I gave a friend an instruction and five inputs. The friend read
the instruction and wrote an output for every one of the inputs.
C. I gave a friend an instruction and five inputs. The friend read
the instruction and wrote an
write the number of positive examples
Therefore, the correct answer is ('positive',

Context 2:

Q. There's a lot to recommend Read My Lips.
A. I gave a friend an instruction and five inputs. The friend read
the instruction and wrote an output for every one of the inputs.
B. I gave a friend an instruction and five inputs. The friend read
the instruction and wrote an output for every one of the inputs.
C. I gave a friend an instruction and five inputs. The friend read
the instruction and wrote an
write the number of positive examples
Therefore, the correct answer is ('positive',
```

L.1.22    SENTENCE SIMILARITY

```
Prompt 0: .

Prompt 1: !:There are two kinds of problems / hints that you will
see:!<

Context 0:

Q. Sentence 1: The polar bear is sliding on the snow. Sentence 2: A
 polar bear is sliding across the snow.
A. : Read the two sentences and decide how well the second sentence
 expresses the same idea as the first. Then enter the number of the
 answer option that best describes your decision.
1. The two sentences express the same idea.
2. The two sentences express the same idea, but the second sentence
 is not as clear as the first.
3. The two sentences express the same idea, but the second sentence
 is
: Given two sentences, your algorithm must determine the similarity
 between the sentences on a scale of 0 to 5. A score of 0 means the
 sentences have nothing in common. A score of 5 means the sentences
 are semantically similar. The
Therefore, the correct answer is 5.

Context 1:

Q. Sentence 1: Iranian President praises nuclear deal Sentence 2:
Iran arrests several spies near nuclear plant
A. . Output the score according to the following rules:
0: The sentences have nothing in common.
1: The sentences are about the same topic, but express different
ideas.
2: The sentences are about the same topic, and express similar
ideas.
3: The sentences are about the
 :
Therefore, the correct answer is 1.

Context 2:

Q. Sentence 1: A girl is playing a flute. Sentence 2: A band is
playing on a stage.
A. .Output the score according to the following rules:
0: The sentences have nothing in common.
1: The sentences are about the same topic, but express different
ideas.
2: The sentences are about the same topic, and express similar
ideas
!: There are two kinds of problems / hints that you will see:!<
Therefore, the correct answer is 1.
```

### L.1.23 WORD IN CONTEXT

```
Prompt 0: Determine whether the given word is used with the same
meaning in both sentences.  Write

Prompt 1: Decide whether the given word is used in the same meaning
in both sentences.

Contexts

Context 0:

Q. Sentence 1: The Times is not the voice of New York. Sentence 2:
The voice of the law. Word: voice
A. Determine whether the given word is used with the same meaning
in both sentences. Writeyes or no.
The answer is yes.
Decide whether the given word is used in the same meaning in both
sentences.
Therefore, the correct answer is (yes).

Context 1:

Q. Sentence 1: Do you communicate well with your advisor? Sentence
2: He and his sons haven't communicated for years. Word:
communicate
A. Determine whether the given word is used with the same meaning
in both sentences. Writeyes or no.
The answer is yes.
Decide whether the given word is used in the same meaning in both
sentences.
Therefore, the correct answer is (yes).

Context 2:

Q. Sentence 1: Can you take me to the main entrance? Sentence 2:
Take a scene. Word: take
A. Determine whether the given word is used with the same meaning
in both sentences. Writeyes or no.
The answer is no.
Decide whether the given word is used in the same meaning in both
sentences.
Therefore, the correct answer is (no).
```

## M    GPT 3.5 RESULTS

We ran PB on GSM-8k using GPT3.5-Turbo-0613, see Figure 5, and GPT-3.5-Turbo-1106, for 8000 and 2500 evaluations respectively. The prompt with the best training set fitness in the run was evaluated on the test set achieving 65.5% and 63.9% respectively. For GPT-3.5-Turbo-0613 the evolved task prompts were: Prompt1: Solve the math word problems and provide the numerical answers without using any words from the original instructions. Prompt2: Solve the math word problems and provide the numerical answers without using any words from the original instructions.. For GPT3.5Turbo-1106 the evolved task prompts were, Prompt1: First, calculate the numerical values of the expressions and arrange them in descending order. Then, provide the final answer in Arabic numeral format as 71. Prompt2: Solve the math word problem, providing the answer in Arabic, while considering diversity and taking breaks as needed.. It is encouraging that Promptbreeder is able to work with very different language models. PS+ prompt (our own implementation) only achieved 44.7% with GPT-3.5-Turbo-0613. In Wang et al. (2023b) PS+ achieved 59.3% 0-shot per-

formance using the text-davinci-003 model. Note GPT 3.5 is reported as getting 57.1% few-shot test set performance on GSM-8k OpenAI (2023), but our evolved results zero shot with GPT3.5-Turbo surpass those by a substantial margin. Table 2 in Wang et al. (2023b) sets expectations with regard to relation between zero-shot and few-shot performance, with few-shot being slightly better-performing than zero-shot. Our zero-shot evaluation demonstrated that the default CoT prompt achieved scores of 52.5% and 53% for GPT-3.5-Turbo-0613 and GPT-3.5-Turbo-1106, respectively. However, a more sophisticated evolved prompt achieved significantly higher scores of 65.5% and 63.9%, respectively.

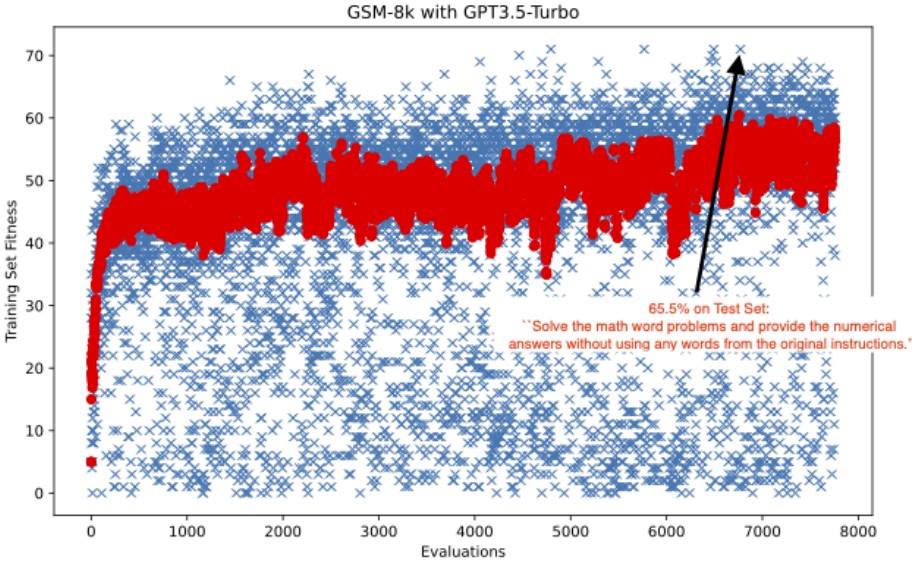

Figure 5: GSM-8k run using GPT-3.5Turbo-0613. Red = mean population fitness, Blue = Individual fitness evaluations. The best evolved prompt on the training set is shown, and its test set fitness is 65.5%

## N    THE EFFECT OF USING POOR PROBLEM DESCRIPTIONS

We looked at the impact of initializing with problem descriptions where it does not accurately describe the problem. Specifically we ran Promptbreeder with the GSM8K dataset using a neutral problem description Write some text and a misleading problem description Write a poem.

### N.1    NEUTRAL PROBLEM DESCRIPTION

The fittest individual at the start of the experiment achieved a test accuracy of 57.5% with the task-prompts 1st part and Trewe sit mtew. After 2.5k evaluations the task-prompts had evolved to frstpart and few it eomt which gets a test accuracy of 66.6%

### N.2    MISLEADING PROBLEM DESCRIPTION

The fittest individual at the start of the experiment achieved a test accuracy of 33.8% with the task-prompts Express yourself in a verse. and Use all you know to help someone. After 2.5k evaluations the task-prompts had evolved to INSTRUCTION MUTANT: Write working out in the answer sheet as well.   and INSTRUCTION: Break new ground and break barriers to innovate with an exciting mutator prompt. resulting in a test accuracy of 57.7%

