# OpenReview forum: "Promptbreeder: Self-Referential Self-Improvement via Prompt Evolution"
_ICLR.cc/2024/Conference — Submitted to ICLR 2024_

### Official Review · Reviewer_AFtS · 2023-10-24

**Soundness:** 2 fair
**Presentation:** 2 fair
**Contribution:** 2 fair
**Rating:** 5
**Confidence:** 5

**Summary:**

This paper proposes Promptbreeder (PB), a general-purpose self-referential self-improvement mechanism of LLMs that evolves and adapts prompts for a given domain.
This mechanism mutates a population of task-prompts, evaluates them for fitness on a training set, and repeats this process over multiple generations to evolve task-prompts.
Authors shows that PB outperforms state-of-the-art prompt strategies such as Chain-of-Thought and Plan-and-Solve Prompting on commonly used arithmetic and commonsense reasoning benchmarks.

**Strengths:**

* This paper focuses on the problems of prompt strategies that hand-crafted prompt-strategies are often sub-optimal,
and present Promptbreeder (PB).
* A self-referential self-improvement mechanism is promissing approach as the prompt optimization.
* The authors conducted  extensively survey and support their originality.

**Weaknesses:**

**The comparative study of alternative methods is weak and does not fully support the validity of the proposed method**
* While Promptbreeder is an important approach as the prompt strategies, it is complex in its composition such as a mutation prompt, a hyper mutation prompt,  a domain-specific problem description, and a seed thinking-styles, and then lacks the ablation analysis to show which components are effective and how effective they are.
* Promptbreeder appears to rely on past prompt strategies or their combination, lacks its motivation and theoretical considerations, and lacks sufficient experimental results to support them.
* As authors use only two LLMs as baselines, the generality of the proposed method cannot be determined.

**Questions:**

* What is the rationale for the baseline selection in Table 1?
* Can you explain the result that PS+ does not show a better performance than PS for PaLM 2-L than text-davinci-003, in Table 1?
* Which resullt supports your claim ``we investigate the various self-referential components of Promptbreeder and their contribution to our results.''?
* Can you show how effective it is compared to LLaMA or OPT as baselines?

---

> ### Author Response · Authors · 2023-11-17
> **Thank you reviewer AFtS**
>
> We thank the reviewer for their detailed comments to improve our paper. We are glad the reviewer found Promptbreeder promising and our survey of related work extensive, supporting the originality of our approach. We address your concerns below.
>
> **Comprehensive Comparative Study:**
> We respectfully disagree with your claim that our paper lacks sufficient experimental results. Promptbreeder was benchmarked against various algorithms across nine distinct problems, in fact going beyond the standards as seen in published state-of-the-art work like APE [1] and Plan-and-Solve [2]. However, to address your point, we've expanded the comparison in Table 1 of our revised draft even further, adding three more control groups to reinforce the validity of Promptbreeder.
>
> [1] Zhou, Y., Muresanu, A. I., Han, Z., Paster, K., Pitis, S., Chan, H., & Ba, J. (2023). Large Language Models Are Human-Level Prompt Engineers. arXiv. https://doi.org/10.48550/arXiv.2211.01910
>
> [2] Wang, L., Xu, W., Lan, Y., Hu, Z., Lan, Y., Lee, R. K.-W., & Lim, E.-P. (2023). Plan-and-Solve Prompting: Improving Zero-Shot Chain-of-Thought Reasoning by Large Language Models. arXiv. https://doi.org/10.48550/arXiv.2305.04091
>
> We also point to the fact that Promptbreeder outperforms APE on 21/24 instruction induction tasks which are outlined in the Appendix L.
>
> **Ablation Analysis:**
>
> You state that our paper “lacks the ablation analysis to show which components are effective and how effective they are”. You will be glad to hear that our original submission already contained a comprehensive ablation analysis in Appendix L, however, we have moved this to Appendix B as it was easily overlooked. It shows that every component of Promptbreeder is important, and the interplay of different mutation operators is what makes Promptbreeder effective. We agree that this ablation analysis is crucial to show which components of Promptbreeder are effective, so to decrease the chances of other readers overlooking this important part of our work, we have made the reference to Appendix B in bold in the main text. We thank you for bringing this to our attention and improving the paper in this way.
>
> **Theoretical Motivation:**
>
> Your observation on Promptbreeder's reliance on past prompt strategies is valuable. We designed Promptbreeder to not only incorporate but also evolve beyond these strategies. The addition of genetic algorithms to our framework is grounded in existing literature, advocating for diverse and non-greedy exploration that circumvents local optima. We've elaborated on this in the revised draft, highlighting how Promptbreeder transcends mere aggregation of existing strategies. Control 1 in our paper demonstrates that initializing task prompts with these strategies is less effective than employing evolutionary evaluations.
>
> **Rationale for Baseline Selection:**
>
> The selection of baselines was driven by a desire to compare Promptbreeder with the latest state-of-the-art prompting strategies. Additionally, two control experiments were conducted to validate the efficacy of evolutionary processes in Promptbreeder over random search and to demonstrate improvements over the initial problem descriptions used as prompts.
>
> **Summary:**
>
> In conclusion, we respectfully ask you to consider the enhancements made in our revised draft, which address your concerns in detail, and to increase your support for our paper.

---

> > ### Author Response · Authors · 2023-11-21
> > **Remaining questions?**
> >
> > Dear reviewer,
> >
> > we believe we have addressed all of your concerns in our rebuttal above. We would love to hear from you whether there is anything that stands in the way for you to increase your support of the paper.

---

> ### Comment · Reviewer_AFtS · 2023-11-22
> **Reply to authors**
>
> Thank you for your answer.
> I found it very interesting.
>
> Could you explain clearly me a little more about the following points?
> I would like to check the effectiveness and novelty of individual proposals.
> * How insensitive is the proposed methodology to existing strategies and types of LLM? Please provide any evidence of this.
> * Also, there are several search methods to prevent genetic algorithms from becoming a local optimum solution, can you compare them and highlight what you have devised for Promptbreeder's?
> * Please tell us about your evaluation methodology for reproducibility. Which tools or library do you use?

---

> > ### Author Response · Authors · 2023-11-22
> > **Thank you**
> >
> > Thank you for the kind words and your additional questions to improve the clarity of our work.
> >
> > **Sensitivity to existing strategies and types of LLM**: We would like to point you to Appendix B (Ablation Analysis) and Appendix M (GPT-3.5 Results). The results in these two appendices demonstrate that Promptbreeder is sensitive to the removal of mutation operators (Appendix B), but it is not sensitive to changing the underlying LLM (Appendix M). That is, Promptbreeder is still effective when replacing PaLM 2-L with GPT-3.5. Thank you for raising this—we will discuss it in more depth in the main paper.
> >
> > **Comparison to genetic algorithms**: In Promptbreeder, we employ several methods to prevent premature convergence. These methods can be compared with traditional search methods in genetic algorithms:
> >
> > 1. **Direct Mutation (Zero-order and First-order Prompt Generation)**: This method in Promptbreeder creates new task prompts either from scratch (zero-order) or by modifying an existing task prompt (first-order). Traditional genetic algorithms often use mutation as a means to introduce variability. Promptbreeder's application of direct mutation serves the same purpose, but it is more linguistically oriented, focusing on generating prompts that are more effective for LLMs.
> >
> > 2. **Estimation of Distribution Mutation**: This approach in Promptbreeder considers patterns in a set of parent prompts to generate new ones. Traditional genetic algorithms might use similar concepts in crossover operations but lack the language focus of Estimation Distribution Mutation in Promptbreeder. Concretely, as this Estimation of Distribution Mutation is driven by an LLM in Promptbreeder, it could exploit linguistic regularities in high-fitness prompts to create even better prompts.
> >
> > 3. **Hypermutation (Zero-order and First-order Hyper-Mutation)**: Hypermutation in Promptbreeder evolves the mutation-prompts themselves. This self-referential approach is unique to Promptbreeder and not typically found in conventional genetic algorithms. It focuses on improving the evolutionary process itself, rather than just the solutions.
> >
> > 4. **Lamarckian Mutation**: This method in Promptbreeder uses successful outcomes to influence the generation of new prompts, effectively learning from past successes. Traditional genetic algorithms don't usually incorporate this "learned experience" directly into the genotype.
> >
> > 5. **Prompt Crossover and Context Shuffling**: These are methods to introduce diversity in Promptbreeder. Crossover in traditional genetic algorithms serves a similar purpose.
> > The key distinction between Promptbreeder and traditional genetic algorithm search methods lies in its focus on evolving language prompts for LLMs, employing linguistically-oriented strategies and a self-referential approach to evolve both the prompts and the method of their evolution. Traditional methods, while also aiming to avoid local optima, do not typically have this dual-layer of evolution nor the specific focus on language and prompt effectiveness.
> >
> > **Tools and Libraries**: Promptbreeder is implemented from scratch in Python, making use of Jax for BERT embeddings in the Emistation of Distribution Mutation, and API calls for interfacing with LLMs—it doesn’t make use of any existing genetic optimization libraries. For evaluation, we use the standard evaluation protocols for the respective benchmarks (MultiArith, SingleEq, AddSub, SVAMP, SQA, CSQA, AQuA-RAT, GSM8K, ETHOS). In Appendix C, D, and E, we have published all initial Thinking Styles, Mutation Prompts, and Task Prompts used in Promptbreeder. This, together with the detailed explanation of Promptbreeder’s mutation operators, should make it straight-forward to reproduce. We will look into open-sourcing Promptbreeder upon acceptance.
> >
> >
> > Thank you again for your constructive feedback to improve our paper. Please let us know if there stands anything else in the way of strengthening your support for our paper.

---

> > > ### Comment · Reviewer_AFtS · 2023-11-23
> > > **Reply to authors**
> > >
> > > Thank you for your answer.
> > > I found it very interesting, too.
> > >
> > > Can you tell me a little more about your evaluation methods and tools?
> > > As we can have available prompt evaluation tools, but have you considered using or modifying these tools?
> > > If so, can you tell us about those tools and the results of your consideration for reproducibility?

---

> > > > ### Author Response · Authors · 2023-11-23
> > > > **Reply**
> > > >
> > > > Thank you for your additional comments. We follow the evaluation protocol and methods of published state-of-the-art prompt strategies such as Chain-of-Thought Prompting [1], Plan-and-Solve Prompting [2], as well as APE [3] and OPRO [4], to name just a few. This is explained in more detail in Appendix J. Note that by evaluating on nine different diverse benchmarks (MultiArith, SingleEq, AddSub, SVAMP, SQA, CSQA, AQuA-RAT, GSM8K, ETHOS) used in prior works we have gone beyond each of the state-of-the-art papers [1-4] which only evaluate on a subset. The evaluation protocol is implemented in Python. We are unsure which available prompt evaluation tools you are referring to—if you could point us to them that would be appreciated so that we can discuss them in the paper. As we have published all initial Thinking Styles, Mutation Prompts, and Task Prompts (Appendix C, D, and E) used in Promptbreeder, provided detailed explanations of Promptbreeder’s mutation operators, and follow standard evaluation protocols as used in prior work [1-4], we believe Promptbreeder is straightforward to reproduce. If you believe something crucial is missing or unclear, please let us know. If you agree that we have addressed your concerns, we would kindly ask you to reconsider your assessment of our work.
> > > >
> > > > [1] Wei, J., Wang, X., Schuurmans, D., Bosma, M., Ichter, B., Xia, F., … Zhou, D. (2023). Chain-of-Thought Prompting Elicits Reasoning in Large Language Models. arXiv. https://doi.org/10.48550/arXiv.2201.11903
> > > >
> > > > [2] Wang, L., Xu, W., Lan, Y., Hu, Z., Lan, Y., Lee, R. K.-W., & Lim, E.-P. (2023). Plan-and-Solve Prompting: Improving Zero-Shot Chain-of-Thought Reasoning by Large Language Models. arXiv. https://doi.org/10.48550/arXiv.2305.04091
> > > >
> > > > [3] Zhou, Y., Muresanu, A. I., Han, Z., Paster, K., Pitis, S., Chan, H., & Ba, J. (2023). Large Language Models Are Human-Level Prompt Engineers. arXiv. https://doi.org/10.48550/arXiv.2211.01910
> > > >
> > > > [4] Yang, C., Wang, X., Lu, Y., Liu, H., Le, Q. V., Zhou, D., & Chen, X. (2023). Large Language Models as Optimizers. arXiv. https://doi.org/10.48550/arXiv.2309.03409

---

### Official Review · Reviewer_DPYS · 2023-10-30

**Soundness:** 2 fair
**Presentation:** 2 fair
**Contribution:** 3 good
**Rating:** 8
**Confidence:** 3

**Summary:**

This paper introduces PromptBreeder (PB), a novel prompt evolution system that automates the exploration of prompts within a specific domain, thereby improving a language model's ability to answer questions in that domain. PB employs a self-referential self-improvement mechanism to evolve and adapt task-prompts. By mutating a population of task-prompts, evaluating their fitness on a training set, and iteratively repeating this process, PB successfully evolves task-prompts. The empirical evidence presented in the paper provides strong support for the effectiveness of PB in enhancing the language model's performance.

**Strengths:**

1. The paper introduces an innovative automatic strategy for prompt discovery, eliminating the requirement for manual engineering and design. This approach streamlines the process and saves time and effort.
2. The proposed prompt strategy showcases exceptional performance when compared to currently available state-of-the-art approaches.
3. The paper is commendable for its clear and well-structured organization. The logical flow of the content enhances readability and comprehension, contributing to a more effective communication of the research findings.

**Weaknesses:**

1. The proposed PB algorithm appears to rely heavily on interactions with the LLM compared to the baselines. As a result, solely evaluating its performance based on accuracy may not provide a fair assessment of its capabilities.
2. While the authors acknowledge that hand-crafted prompt-strategies are often sub-optimal, they do not offer a guarantee or highlight any asymptotic properties for the PB algorithm, leaving room for uncertainty regarding its long-term effectiveness.
3. The main text of the paper is relatively concise, with several crucial aspects relegated to the appendix. This arrangement can disrupt the smoothness of the reading experience.

**Questions:**

1. I am quite curious about the sample efficiency of the algorithm. As evolutionary algorithms often suffer from the poor sample efficiency.
2. I hope that authors will provide how PB perforems if the size of train dataset is limited.
3. It will be nice to provide guarantee or asymptotic property for the PB.

---

> ### Author Response · Authors · 2023-11-17
> **Thank you reviewer DPYS**
>
> We thank the reviewer for their insightful feedback and support for our paper. We are glad to hear they found the method innovative, the empirical performance exceptional, and the write up clear and well structured. We address your concerns below.
>
> **Reliance on LLM Interactions:**
>
> We appreciate your point about Promptbreeder’s interaction intensity with LLMs. As a general-purpose self-referential self-improvement mechanism, Promptbreeder leverages LLMs to mutate a population of task-prompts, evaluating their fitness on a training set over multiple generations. This process inherently involves significant interaction with LLMs, but it is crucial for the evolutionary approach that underpins Promptbreeder's effectiveness​​. The sample efficiencies of the algorithm for each dataset are shown in the Appendix K for each problem. For example, on AQuA the best individual was produced after 1400 mutations which equates to 1400 x 2 x 100 Q:A pairs tested. While this may seem costly at first, we believe Promptbreeder will have important practical implications as the time and cost to find a superior prompt is a one-time cost for a given domain, and amortized with respect to many LLM inferences at deployment time of the prompt.
>
> **Lack of Asymptotic Guarantees:**
>
> Note that such guarantees are generally not available for evolutionary algorithms in realistic large and noisy problem spaces. However, we have demonstrated empirical superiority to other optimization techniques such as APE, random task prompt initialization (see our own controls in Table 1), and ORPO on GSM-8k. We have shown that using the diverse set of initial task prompts improves Promptbreeder’s search for effective prompts in various domains.
>
> **Content Distribution in Main Text and Appendix:**
>
> Thank you for your suggestion to improve our paper. We have moved Figure 3 into the main paper in improved form.
>
>
> Again, thank you for your valuable contribution to improving our paper.

---

### Official Review · Reviewer_ceRr · 2023-11-01

**Soundness:** 3 good
**Presentation:** 3 good
**Contribution:** 2 fair
**Rating:** 6
**Confidence:** 3

**Summary:**

This paper proposes Promptbreeder, a self-referential self-improvement method to evolve prompts for a specific domain. Given some seed prompts, domain description, and thinking-styles, Promptbreeder can generate variations of both task prompts and mutation prompts.  Experiments on various benchmarks have verified that the method outperforms other prompt strategies like CoT and Plan&Solve.

**Strengths:**

This paper proposes a systematic framework to evolve domain-specific prompts, and shows better results compared to other prompt strategies.

**Weaknesses:**

1. The experiment is not extensive. In Table 1, the compared LLMs do not involve the most recognized models like gpt-3.5 or gpt-4, and the compared methods should contain CoT on PaLM 2-L.
2. The proposed method Promptbreeder still requires initial information for specific task (like description or mutation prompts), where worse initialization may lead to worse performance. This makes the method may not generalize to various tasks.

**Questions:**

None

---

> ### Author Response · Authors · 2023-11-17
> **Thank you reviewer ceRr**
>
> Thank you for your thoughtful review of our paper on Promptbreeder. We appreciate your recognition of our systematic framework's ability to evolve domain-specific prompts and its comparative effectiveness. We address your concerns below and offer clarifications and updates to our methodology and results.
>
> **Regarding the Experiment's Extensiveness:**
>
> We acknowledge your observation about the lack of inclusion of prominent models like GPT-3.5 or GPT-4 in our initial experiments. In response, we have carried out experiments with GPT-3.5 on GSM-8k showing successful improvement of prompts to 65.5% test set accuracy, see Appendix M for further details. Additionally, as per your suggestion, we have re-run the experiments using Chain-of-Thought (CoT) and other methods on PaLM 2-L. Our updated results demonstrate that Promptbreeder shows consistent improvements.
>
> **Concerning the Requirement of Initial Information:**
>
> You rightly pointed out that the effectiveness of Promptbreeder depends on the initial information provided, such as task descriptions or mutation prompts. To address this, we have included a new section in the Appendix of our revised draft and carried out two more control experiments showing that even with poor and general problem descriptions, PB shows consistent improvements in fitness.
>
> - **Generation of Thinking Styles and Mutation Prompts from Problem Descriptions:** We have developed a method by which thinking styles and mutation prompts can be autonomously generated from the problem description itself, reducing the dependency on external initialization.
> - **Concerns about the initial problem description information provided to Promptbreeder:** We demonstrate (Appendix N) that even with neutral or misleading problem specification (task descriptions) Promptbreeder achieves substantial improvement in terms of the test set fitness of evolved prompts.
>
> We respectfully disagree with the implication that our method may not generalize across various tasks. The aforementioned enhancements and experimental results strongly suggest that Promptbreeder is not only adaptable but also capable of performing effectively across a diverse range of tasks, even with minimal or suboptimal initial information. Furthermore, the fact that Promptbreeder outperforms APE on 21/24 instruction induction tasks also highlights this, see Appendix L.
>
> **Summary:**
>
> We believe these updates and clarifications address your concerns regarding the breadth of our experiments and the generalizability of Promptbreeder. We kindly request you to reconsider the contribution score in light of these significant improvements and additional data. Moreover, if there are any further queries or points of contention, we are more than willing to provide additional clarifications to strengthen the case for acceptance of our paper.

---

> > ### Author Response · Authors · 2023-11-21
> > **Remaining questions?**
> >
> > Dear reviewer,
> >
> > we believe we have addressed all of your concerns in our rebuttal above. We would love to hear from you whether there is anything that stands in the way for you to increase your support of the paper.

---

> > > ### Comment · Reviewer_ceRr · 2023-11-23
> > > **Thanks for the Responses**
> > >
> > > I have carefully read the responses from authors, they have addressed most of my concerns. I am willing to increasing my score to 6. Thanks!

---

### Official Review · Reviewer_7CXL · 2023-11-10

**Soundness:** 2 fair
**Presentation:** 2 fair
**Contribution:** 2 fair
**Rating:** 5
**Confidence:** 2

**Summary:**

This manuscript proposed PROMPTBREEDER, a new self-referential self-improvement method using an LLM to generate and refine both task- and mutation- prompts over multiple generations. PROMPTBREEDER incorporated nine mutation operators falling into five broad classes to promote varied and robust prompt evolution. This method has been evaluated in 8 tasks with promising performance using the PaLM 2-L model, surpassing serval established baselines such as CoT, APE, and Plan-and-Solve.

**Strengths:**

- PROMPTBREEDER showed promising performance and outperformed competing baseline approaches in 7 out of 8 tasks with a large margin.
- The PROMPTBREEDER method was well elaborated in the manuscript and in the supplementary material.

**Weaknesses:**

- PROMPTBREEDER aimed to concurrently refine both task and mutation prompts, considerably expanding the search space. However, the absence of navigation during each evaluation often resulted in unpredictable performance for the successive generated prompts. This is evidenced by the persistence of less effective prompts after extensive evaluations, as illustrated in Figure 3.
- In light of the above, PROMPTBREEDER appears to rely on an extensive series of trial-and-error iterations to identify an optimized prompt, raising concerns about the method's efficiency in exploring potential solutions. It would be helpful if the authors can include a comparative analysis detailing the correlation between the number of prompts generated and the performance for each evaluated baseline and for PROMPTBREEDER itself.
- It is not clear how the "Mutator Prompts" (Table 2) and "Thanking Styles" (Section D) are created. Are they derived from pre-existing prompt strategies? Are these prompts hand-crafted?

**Questions:**

1. This study exclusively shows the performance of PROMPTBREEDER with PaLM 2-L, raising questions about its generalization ability to other LLMs. Specifically,

- Whether PROMPTBREEDER method can be effectively utilized to enhance prompts for LLMs?

- Whether the prompts refined using PROMPTBREEDER in conjunction with PaLM 2-L can yield improved results when employed with alternative LLMs.

2. In Figure 3, the y-axis label is not visible. Additionally, what is the relationship between "number of evaluations" and "number of generations"? It is confusing since Section 4 reported that the populations "evolved for typically 20-30 generations," but there are 2000 evaluations in Figure 3.

3. Please clarify why OPRO was only evaluated on GSM8K in Table 1.

Minor Comment:

For the sake of readability, it would be beneficial if the color coding is consistent across different figures and texts. For example, the "mutation prompt" is color-coded as red in the text on Page 6, yet appears in shades of blue (and not the exact same blue) in Figures 1 and 2.

---

> ### Author Response · Authors · 2023-11-17
> **Thank you reviewer 7CXL**
>
> We thank the reviewer for their detailed comments to improve our paper. We are glad to hear the reviewer found our empirical results promising and the method well elaborated. Below, we are addressing the reviewer’s concerns.
>
>
> **Generalization:**
>
> Many thanks for the expressing the concern that this study exclusively shows the performance of PROMPTBREEDER with PaLM 2-L. To address this we ran PB on GSM-8k using GPT3.5-Turbo achieving test set scores of 65.5% and  63.9% accuracy respectively. Results are shown in Appendix M. The PS+ prompt only achieved 44.7% with GPT-3.5-Turbo. This provides evidence that PB generalizes to other language models.
>
> **Absence of Navigation:**
>
> The absence of directed navigation is an important component of an evolutionary algorithm. The success of evolution relies on sequences of random mutations that, when accumulating over time, lead to increased fitness. Promptbreeder, an evolutionary algorithm, makes use of this property. In fact, even keeping around less fit (but less specialized) individuals in an evolving population can be important for the entire evolutionary system to arrive at better solutions in the long term [1]. A more directed Promptbreeder approach is conceivable, but it would move Promptbreeder away from an evolutionary approach which is outside of the scope of our paper.
>
> [1] Sudholt, D. The Benefits of Population Diversity in Evolutionary Algorithms: A Survey of Rigorous Runtime Analyses, 2018. https://arxiv.org/abs/1801.10087
>
> **Efficiency in Prompt Optimization:**
>
> In response to your concern about efficiency, we have introduced two control experiments: the Random Baseline and the PD baseline. The Random baseline evaluates 2,000 prompts generated by our initialization method without evolution, demonstrating that evolution contributes more than a random search. The PD baseline, where task-prompts are set to the problem description, shows that evolution improves performance beyond the initial problem description.
>
> **Origin of "Mutation Prompts" and "Thinking Styles":**
>
> We appreciate your inquiry about the creation of the "Mutation Prompts" and "Thinking Styles." These were hand-designed for general problem-solving domains and are thus widely applicable (as demonstrated by our empirical gains on multiple diverse reasoning benchmarks). We have added a new section (Appendix G) detailing how an LLM can generate these lists from a problem description using a hierarchical generation method.
>
> **Transfer of Promptbreeder Prompts to other LLMs:**
>
> To address your question about Promptbreeder's applicability to other LLMs, note that we do not claim prompt transferability across LLMs—Promptbreeder is designed to find prompts that are tailored to the unique characteristics of each LLM for optimal task performance through automated prompt engineering. We will clarify this in a revised version of the paper.
>
> **Clarifications on Figure 3, OPRO Evaluation, and Color Coding:**
>
> We have corrected and enhanced Figure 3, providing clarity on the relationship between evaluations and generations (a generation being POP_SIZE evaluations). Regarding OPRO's evaluation on GSM8K, we suggest the authors of OPRO may consider evaluating it on a broader range of tasks, as we have done in our study. We have standardized the color coding for task-prompts and mutation-prompts throughout the paper to enhance readability and coherence.
>
> **Summary:**
>
> We again thank the reviewer for helping improve our paper. We hope that our clarifications will lead you to consider increasing your rating of our paper or detailing what still stands in the way of increasing your support.

---

> > ### Author Response · Authors · 2023-11-21
> > **Remaining questions?**
> >
> > Dear reviewer,
> >
> > we believe we have addressed all of your concerns in our rebuttal above. We would love to hear from you whether there is anything that stands in the way for you to increase your support of the paper.

---

> ### Comment · Reviewer_7CXL · 2023-11-22
> **Thanks for the responses**
>
> Thank you for your thorough responses, which have addressed many of my initial concerns through additional experimental results and analysis.
>
> However, I continue to have reservations about the *efficiency* of the proposed PB. Although the use of an evolutionary algorithm demonstrates certain advantages over the '2000 random prompts' approach, the marginal improvement of PB over the Random baseline is less pronounced when compared to other prompting strategies except in the GSM8K. As evidenced in Table 1, the Random baseline outperforms most other baselines in 7/8 datasets, suggesting that a large number of various prompts plays a more significant role in most tasks, which lacks novelty, in my opinion.
>
> Moreover, I am concerned about the *scalability* of experiments using PB, especially when 2k prompts are needed to be evaluated. Factors such as dataset size, input length, and the use of different LLMs when opting for paid APIs like GPT-4 over GPT-3.5-Turbo, could substantially increase the time and cost of experimentation. This could pose a disadvantage for researchers and practitioners with limited resources.
>
> In conclusion, while the revised manuscript shows commendable improvements and the authors have provided detailed clarifications, the persistent concerns mentioned above led me to **maintain my initial rating**.

---

> > ### Author Response · Authors · 2023-11-23
> > **Response**
> >
> > Thank you for engaging with our rebuttal and recognizing commendable improvements, detailed clarifications, and resolving many of your initial concerns.
> >
> > **Efficiency**: We respectfully but firmly disagree with your view that Promptbreeder’s success can be mainly attributed to the initial set of prompts (random baselines). While you are correct that this initial set of prompts outperforms Plan-and-Solve on 7 out of 8 benchmarks, it is important to note that Promptbreeder outperforms this random baseline on 8 out of 8 benchmarks, with notable improvement on AddSub (+2.0pp), SVAMP (+3.2pp), CSQA (+3.5pp), AQuA-RAT (+4.3pp) and GSM8K (+18.4pp). Expecting Promptbreeder to provide drastic improvements (like the +18.4pp on GSM8K) over strong baselines across the board is, in our view, setting completely unrealistic expectations. This is in particular true when baselines already start off at a high accuracy on a benchmark. For example, take Promptbreeder’s improvement from 99.4% to 99.7% over the baseline on the MultiArith benchmark. This corresponds to an error reduction by 50% which would be equivalent to +17.25pp on GSM8K where the baseline starts at an accuracy of 65.5%. Furthermore, the ablation analysis in Appendix B clearly demonstrates that each component in Promptbreeder is crucial to improve performance across a wide range of reasoning benchmarks. If only initial prompts mattered, we wouldn’t see a clear drop in performance when ablating hyper-mutation or Lamarckian mutation operators.
> >
> > **Scalability**: We acknowledge your concern regarding the cost of running Promptbreeder with paid APIs like GPT-3.5 or GPT-4. Our new results presented in Appendix M on improving GPT-3.5 reasoning abilities on GSM8K using Promptbreeder cost $300 in total. While this is not extremely cheap, we believe Promptbreeder will have important practical implications as the time and cost to find a superior prompt is a one-time cost for a given domain, and amortized with respect to many LLM inferences at deployment time of the prompt. We don't follow your argument that our paper would disadvantage researchers and practitioners with limited resources. Our paper is an existence proof and we are confident that future work by researchers and practitioners will further improve upon it and make it more scalable, in particular given the continuing trend of compute becoming cheaper.
> >
> > We hope these further clarifications convinced you to reconsider your assessment of our work and to improve your rating.

---

### Official Review · Reviewer_PsCD · 2023-11-10

**Soundness:** 3 good
**Presentation:** 3 good
**Contribution:** 3 good
**Rating:** 5
**Confidence:** 3

**Summary:**

In this paper, authors propose a genetic algorithm to improve both task prompts and additional “mutation” prompts for the specific downstream task. In details, authors utilize direct mutation, estimation of distribution mutation, hypermutation, Lamaerckian mutation and prompt crossover and context shuffling to randomly change either only task prompt or both task prompt and mutation prompts. The overall genetic algorithm is based on a binary tournament genetic algorithm framework where two individuals are sampled and the worse one is replaced with the mutated version of the better one. In most experiments, authors show that their method achieved better results in the targeted downstream tasks.

**Strengths:**

1: How to generate appropriate LLM prompt for the target downstream task is indeed a very important research question and still lacks a solid answer. I agree with authors that LLM are qualitiifly different from other deep learning models as they have the potential to self-improve their thinking process (e.g. self-generating prompts).

2: From the point of view of genetic algorithm, authors propose a comprehensive set of mutation stratergy, which includes certain extent of "self-referential'/self-improvement mutation stratergy. Overall it seems interesting in general.

**Weaknesses:**

1: The major concern I have is whether evolution algorithm framework in general is not capable enough for the large prompt space for LLM. Overall from the examples provided by the authors in Figure3 appear to show not much different between prompts, which might suggest under-explored prompt space. Personally I feel certain level learning/gradient signal is needed to better explore and generate the prompts for complex LLM models. It will be very interesting (but not necessay) to have some comparision with prompt tunning algorithms if white box LLM models are used.

2: Some result sections in appendix should be moved to the main text as it really helps to show how the algorithm improve the prompts and how the prompts look in the end.

3: I am afraid that I am not familiar with evolution algorithm literature but I personally feel the overall novel comtribution of this paper is limited as it seems all mutation operators are pretty standard, even those "self-referential" ones. And the backbone evolution algorithim seems very simple and out of box.

4: A minor point is that in the result tables, half of the baselines are using different LLM models, which are not directly comparable to authors' method. I strongly encourage authors to rerun the baselines with the same models if possible, or just remove them from the table.

**Questions:**

Please see my comments in weakness sections.

Overall, my main question is how such method will compare with prompt tunning method?

---

> ### Author Response · Authors · 2023-11-17
> **Thank you reviewer PsCD**
>
> We thank the reviewer for their detailed comments to improve our paper. We are glad to hear the reviewer agrees that Promptbreeder tackles the very important research question of whether LLMs can self-improve their “thinking process” via self-generating prompts. Furthermore, it is great to hear that the reviewer found the set of mutation strategies investigated in this work comprehensive, and the paper generally interesting. Below, we are addressing the reviewer’s concerns.
>
> **1.& 2. Prompt Space Exploration in Evolution Algorithms:**
>
> We acknowledge your concern regarding the adequacy of evolution algorithms for exploring the large prompt space in Large Language Models (LLMs). To address this, we have updated Figure 3 (and, as suggested, moved it from the Appendix to the main part of the paper—thank you for the suggestion!) to showcase not only the highest-performing ('elite') prompts but also a variety of low-fitness prompts generated during the exploration phase. This figure shows that there is in fact considerable exploration in prompt space, with many prompts being produced that are very different to the elite prompt. Elite prompts also change in subtle ways throughout the evolutionary run and achieve much higher fitness at the end. The final prompt is "Sentences are given, and a single word. The output should indicate whether the given word has the same sense in the two given sentences, yes or no." with an early elite prompt in the run being "I'll give you two sentences and a word. Your task is to write if the meaning of the word is the same in both sentences or not."
>
> **1. Comparison with Gradient-Based Prompt Tuning (i.e. Soft Prompting) Methods:**
>
> While we agree that a comparison to soft-prompting methods could be valuable, it falls outside of the scope of this work as well as outside of the scope of prior published state-of-the-art prompt strategies such as Chain-of-Thought Prompting [1], Plan-and-Solve Prompting [2], as well as APE [3] and OPRO [4], to name just a few.
>
> [1] Wei, J., Wang, X., Schuurmans, D., Bosma, M., Ichter, B., Xia, F., … Zhou, D. (2023). Chain-of-Thought Prompting Elicits Reasoning in Large Language Models. arXiv. https://doi.org/10.48550/arXiv.2201.11903
>
> [2] Wang, L., Xu, W., Lan, Y., Hu, Z., Lan, Y., Lee, R. K.-W., & Lim, E.-P. (2023). Plan-and-Solve Prompting: Improving Zero-Shot Chain-of-Thought Reasoning by Large Language Models. arXiv. https://doi.org/10.48550/arXiv.2305.04091
>
> [3] Zhou, Y., Muresanu, A. I., Han, Z., Paster, K., Pitis, S., Chan, H., & Ba, J. (2023). Large Language Models Are Human-Level Prompt Engineers. arXiv. https://doi.org/10.48550/arXiv.2211.01910
>
> [4] Yang, C., Wang, X., Lu, Y., Liu, H., Le, Q. V., Zhou, D., & Chen, X. (2023). Large Language Models as Optimizers. arXiv. https://doi.org/10.48550/arXiv.2309.03409
>
> **3. Simplicity and Novelty of Mutation Operators:**
>
> Although some of the mutation operators in Promptbreeder are standard in evolutionary algorithms, their application within the domain of LLMs is novel. Also some operators simply cannot be implemented without LLMs, e.g. the lineage based operator, or the context to prompt operator. Designing these operators for LLMs posed unique challenges, particularly in balancing diversity and relevance of the generated prompts. We demonstrate the effective use of LLMs in evolutionary algorithms, a contribution we believe is significant in this field. Furthermore, we argue that the simplicity of the individual mutation operators of Promptbreeder is in fact a strength of the approach, demonstrating that existing mutation operators can be orchestrated to create a capable self-referential self-improving system.
>
> **4. Base LLM comparison:**
>
> Thank you for this comment. We have extended the baselines with PaLM-2L as suggested, and added Chain-of-Thought Prompting (COT and Manual COT) with PaLM-2L as the base model to the results. As expected, Promptbreeder outperforms these baselines by a large margin.
>
>
> **Summary:**
>
> We hope that our clarifications will lead you to consider increasing your rating of our paper or detailing what still stands in the way of increasing your support, so that we may further improve it.

---

> > ### Author Response · Authors · 2023-11-21
> > **Remaining questions?**
> >
> > Dear reviewer,
> >
> > we believe we have addressed all of your concerns in our rebuttal above. We would love to hear from you whether there is anything that stands in the way for you to increase your support of the paper.

---

### Author Response · Authors · 2023-11-18
**Updates to paper draft in response to reviewer feedback**

Dear Reviewers

In response to the feedback from multiple reviewers, we have made significant updates and clarifications in our revised draft. These changes aim to address the concerns raised, and to enhance the overall quality and clarity of our work. The following changes have been made to the paper:

1. Two control experiments were carried out and results added as rows to Table 1: one comparing random search vs. evolution, the other looking at the test set performance using only the original problem description used as a task prompt directly. Promptbreeder improves upon both controls. [**DPYS, PsCD, 7CXL**]
2. Figure 3 moved from appendix to main paper and improved to show improvement of prompts and diversity of the evolutionary exploration. [**PsCD, 7CXL, DPYS**]
3. CoT and Manual CoT prompt were tried with PALM-2-L and added to Table 1. This baseline is inferior to Promptbreeder. [**PsCD, ceRr**]
4. A method for creating thinking styles and mutation prompts from the problem description itself has been added to Appendix G. [**ceRr, 7CXL**]
5. We made the coloring of prompt types consistent throughout the paper. [**7CXL**]
6. Ablation results moved to an earlier section in appendix B immediately after the glossary and highlighted in the results section of the main text for improved prominence. [**AFtS**]
7. We made the rationale for baselines in Table 1 clearer, and explained the rows in more detail. [**AFtS**]
8. We ran Promptbreeder on GSM-8k using GPT3.5-Turbo achieving test set scores of 65.5% and  63.9% accuracy respectively. Results are shown in Appendix M. The PS+ prompt only achieved 44.7% test set accuracy with GPT-3.5-Turbo. This demonstrates that Promptbreeder can be readily applied to other LLMs and is effective there as well. [**ceRr, 7CXL**]
9. We demonstrate in Appendix N prompt improvement even when starting with poor or even completely misleading problem descriptions. [**ceRr**]

We thank the reviewers for their help in improving the paper and hope they consider strengthening the support for our work.

---

### Public Comment · ~Sam_Lee1 · 2023-11-19
**Main text over 9-page limit**

It seems the main text is 10 pages, over the 9-page limit. Is this allowed?

---

> ### Author Response · Authors · 2023-11-19
> **Original submission is 9 pages**
>
> Hi Sam,
>
> Our original submission is 9 pages. We only went to 10 pages during the rebuttal phase to address the reviewer's comments. Multiple reviewers asked us to move content from the appendix to the main part of the paper.
>
> Kind regards,
> Authors

---

> > ### Public Comment · ~Sam_Lee1 · 2023-11-19
> > **Page limit applies to both the initial and final camera ready version**
> >
> > The Call For Papers says:
> >
> > "There will be a strict upper limit of 9 pages for the main text of the submission, with unlimited additional pages for citations. This page limit applies to both the initial and final camera ready version."
> >
> > I suppose the 9-page limit always apply?

---

> > > ### Author Response · Authors · 2023-11-20
> > > **Thanks**
> > >
> > > Hi Sam,
> > >
> > > Thanks for flagging this. Yes, it seems we will have to cut it down to 9 pages again for a potential camera ready. Note that the 10th page predominantly comes from having moved Figure 3 from the Appendix to the main part of the paper as requested by reviewers **PsCD**, **7CXL**, and **DPYS**. We can move it back to the Appendix, but to not confuse reviewers we are going to await feedback by reviewers and guidance by the AC before making any further changes.
> > >
> > > Kind regards, Authors

---

> > > > ### Comment · Program_Chairs · 2023-11-22
> > > > **Please keep revision to 9 pages.**
> > > >
> > > > It is fine to re-order the appendix to highlight the additional figure first. But in light with having a consistent policy for managing 7K submissions, please keep the main paper in the revision to 9 pages.
> > > >
> > > > ICLR Program Chairs

---

> > > > > ### Author Response · Authors · 2023-11-22
> > > > > **Revised back to 9 pages**
> > > > >
> > > > > We have uploaded a version that is 9 pages. We have moved Figure 3 back into the Appendix. To have the Appendix letters be consistent with our rebuttal and to not confuse reviewers, Figure 3 can be found in **Appendix α** at the front of the Appendices.
> > > > >
> > > > > Thank you,
> > > > > Authors

---

### Meta-Review · Area_Chair_TAjV · 2023-12-07

**Metareview:**

This paper introduces Promptbreeder (PB), a self-referential self-improvement method to evolve prompts for a given domain. Given some initial prompts, domain description, and thinking styles, Promptbreeder mutates a population of task-prompts and mutation-prompts, where the mutation of task-prompts is governed by mutation-prompts. Experiments on various benchmarks and Large Language Models show that the proposed method outperforms other prompt strategies such as CoT and Plan and Solve.

Currently, this paper has the following major weakness that has not been addressed during the rebuttal.

As the proposed method is based on evolution algorithms that suffer from sample inefficiency, evaluating it against gradient-based prompt tuning is necessary, but currently absent. This inefficiency of the proposed method is also evidenced as PB only has a marginal improvement over the Random baseline. Thus, the efficiency of the proposed PB is inconclusive.

I suggest the paper should add more baselines to verify the efficiency of the proposed method and resubmit it to other ML venues.

**Justification For Why Not Higher Score:**

The proposed method is based on evolution algorithms that suffer from sample inefficiency. The paper does not give enough evidence to support this is not the case.

**Justification For Why Not Lower Score:**

N/A

---

### Decision · Program_Chairs · 2024-01-16

Reject